# Spike Camera Autofocus via Frequency-Domain Spectral-Centroid Migration

**Xijie Xiang** [1]   **Lin Zhu** [2]   **Yonghong Tian** [1]

## Abstract

Autofocus for spike cameras is challenging because their sparse binary measurements do not provide reliable instantaneous gradients, and noise or illumination drift often breaks the unimodal assumptions behind conventional focus measures. We show that during a focus sweep, the stable sensor-observable cue is a persistent *migration of spectral energy* in the frequency domain: energy shifts outward toward higher frequencies when approaching focus and recedes under renewed defocus. Building on this observation, we propose **CEN** (*Centroid-based Energy Navigation*), a frequency-domain autofocus method that measures spectral migration via a bounded spectral centroid computed on accumulated spike blocks, without image reconstruction or explicit edge extraction. To handle multi-peak and irregular responses in real scenes, CEN further performs *structure-consistent* response identification, selecting the frequency bound whose curve exhibits a clear, localized, interior extremum, followed by robust peak localization using a weighted near-maximum centroid. Experiments on spike-camera dataset demonstrate that CEN achieves the best overall accuracy and response discriminability across diverse scenes, motion types, and illumination variation patterns.

## 1. Introduction

Autofocus is a fundamental component in computational imaging systems (Najibi et al., 2019), enabling the acquisition of sharp visual information under varying focus conditions and optical settings (Firestone et al., 1991). While a wide range of autofocus methods have been developed for conventional frame-based cameras (Zhang et al., 2018), their underlying assumptions are closely tied to the availability of dense and continuous intensity measurements (Shih,

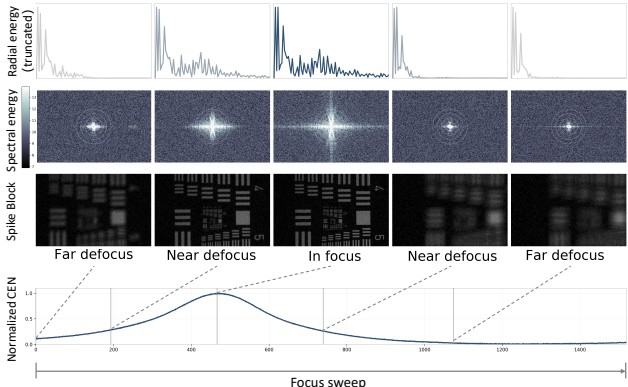

*Figure 1.* During a focus sweep in spike-camera autofocus, focus variation induces a persistent migration of spectral energy in the frequency domain. As the lens moves toward focus, spectral energy shifts toward higher frequencies, increasing the spectral centroid (CEN); after passing the focal plane, it recedes toward lower frequencies. The figure visualizes this outward-and-return CEN migration using representative accumulated spike blocks, their log power spectra, truncated radial energy distributions, and the resulting spectral-centroid response over the sweep. This spectral-centroid migration provides a stable and physically meaningful cue for focus estimation in spike cameras.

2007). As emerging neuromorphic sensors (Gallego et al., 2020), such as spike cameras (Dong et al., 2019), depart significantly from this imaging paradigm, designing effective autofocus mechanisms for spike-based vision remains an open and challenging problem.

Spike cameras encode visual information as sparse binary spikes triggered by threshold crossings of photoreceptor responses. Although the sensing mechanism is driven by absolute light intensity, the output is quantized and temporally sparse, which fundamentally alters the form of observable information. As a result, spatial gradients and fine-scale intensity variations cannot be reliably estimated from a single time instant, and noise manifests in a highly non-Gaussian and scene-dependent manner. Consequently, autofocus criteria originally designed for images, such as gradient-based sharpness or contrast measures, often fail to produce stable or interpretable responses when directly applied to spikes.

To date, spike-camera autofocus has been explored almost exclusively through the spike dispersion (SD) (Su et al., 2024), which evaluates focus by scoring accumulated spike responses along a focus sweep and selecting the maximum response as the focus estimate. In practice, however, spike-

Code and dataset: https://github.com/XIJIE-XIANG/CEN
[1]Peking University [2]Beijing Normal University. Correspondence to: Yonghong Tian <yhtian@pku.edu.cn>, Lin Zhu <linzhu@bnu.edu.cn>.

based focus responses can deviate from simple unimodal behavior, especially in long sequences or dynamic scenes, where multiple local extrema or irregular fluctuations may arise, making focus localization more challenging.

In this work, we revisit the autofocus problem from a different perspective by examining what information is fundamentally observable in spike measurements during focus sweeping. Through extensive empirical analysis, we observe that while spatial-domain sharpness is not directly accessible, focus variation induces a consistent migration of spectral energy in the spatial frequency domain. As the lens moves from defocus toward focus, spectral energy shifts toward higher spatial frequencies, and recedes toward lower frequencies as defocus increases again. This frequency-domain migration provides a stable and physically meaningful cue for focus estimation in spike cameras. Figure 1 illustrates this phenomenon and its associated spectral-centroid response over a focus sweep, suggesting that focus variation can be reliably characterized in the frequency domain.

Motivated by this observation, we formulate autofocus as a frequency-domain statistical problem and propose a spectral centroid–based focus measure (CEN) tailored to spike data. By tracking the center of mass of spectral energy within a controlled frequency range, the proposed measure captures the dominant frequency shift induced by focus variation without requiring intensity reconstruction or explicit edge extraction. Importantly, this formulation makes minimal assumptions about the spatial structure of the scene and is naturally compatible with the sparse and binary nature of spike measurements.

Nevertheless, due to sensor noise, scene dynamics, and the intrinsic sparsity of spikes, the spectral centroid response does not always present a clearly distinguishable focus cue in real-world scenarios. We introduce a structure-consistent response identification that identifies the frequency band and response segment most consistent with the expected focus-induced spectral migration. This strategy enables reliable focus estimation even when the global response contains multiple local extrema or irregular fluctuations.

In summary, this paper makes the following contributions:

1. We present a frequency-domain formulation of autofocus specifically designed for spike cameras, revealing focus-induced spectral energy migration as a robust and interpretable cue.
2. We propose a spectral centroid-based focus measure that is naturally compatible with the sparse and binary nature of spikes, without relying on image reconstruction or spatial-domain sharpness metrics.
3. We introduce a structure-consistent response identification that enables reliable focus estimation without relying on explicit or idealized response patterns, significantly improving robustness across diverse scenes.

## 2. Related work

**Autofocus in frame-based imaging.** Conventional autofocus methods (Najibi et al., 2019; Zhang et al., 2018; Wang et al., 2021) typically evaluate focus quality along a focus sweep (Zhang et al., 2024; Zhou et al., 2012; Subbarao & Tyan, 1998) using image-based focus measures, including spatial sharpness operators (e.g., gradient (DiMeo et al., 2021) and Laplacian (Chantara & Ho, 2016)), statistical contrast criteria, and transform-domain measures (Chantara & Jeon, 2019) based on DCT (Zhang et al., 2016), FFT (Li et al., 2001), or wavelets (Xie et al., 2007). These measures are widely studied in contrast-detection autofocus (Chen & van Beek, 2015; Śliwiński et al., 2019), shape-from-focus (Nayar & Nakagawa, 1994; Pertuz et al., 2013b), and microscopy (Firestone et al., 1991; Pinkard et al., 2019; Hsu et al., 2009), with surveys and comparative evaluations summarizing their behavior and limitations (Shih, 2007; Santos et al., 1997). They generally assume that focus-related statistics are reliably observable from intensity images at each focus position (Shih, 2007).

**Neuromorphic autofocus.** Neuromorphic vision sensors (Gallego et al., 2020), including event cameras and spike cameras, produce sparse measurements without dense intensity frames, motivating autofocus cues that are directly sensor-observable. For event cameras (Lichtsteiner et al., 2008; Brandli et al., 2014), representative cues include event rate (Lin et al., 2022), polarity symmetry (Bao et al., 2023; Ge et al., 2023), and illumination-robust statistics (Qu et al., 2024), often paired with efficient search schemes. Related ideas also appear in all-in-focus reconstruction (Lou et al., 2023; Teng et al., 2024), defocus-aware restoration (Zhu et al., 2025), and depth-from-focus estimation (Jiang et al., 2024; Haessig et al., 2019). For spike cameras, spike dispersion (SD) (Su et al., 2024) evaluates focus using temporal spike statistics on accumulated blocks.

**Reconstruction-based and focal-stack approaches.** Another direction reconstructs intensity-like images from neuromorphic or spike measurements (Dong et al., 2019; Zhu et al., 2022; Xiang et al., 2023) and then applies classical image-based autofocus criteria. Focal-stack and all-in-focus pipelines similarly fuse multiple focus slices to extend depth of field (Teng et al., 2024). While effective in some settings, these approaches introduce additional reconstruction stages and artifacts, and their autofocus performance depends on reconstruction reliability.

**Positioning of our work.** Our contribution is to first identify a spike-observable focus cue and reformulate spike-camera autofocus as a frequency-domain representation problem. By making focus-induced spectral energy migration measurable through a bounded spectral centroid, CEN enables frequency-domain autofocus directly from spike data, without intensity reconstruction or spatial-domain sharpness estimation.

# 3. Method

We address spike-camera autofocus from the perspective of *what focus variation makes observable* in sparse binary spikes. Unlike frame-based imaging, spike data do not provide reliable instantaneous spatial gradients, nor stable spatial-domain sharpness cues. Instead, as established in the previous section, focus sweeping induces a consistent *migration of spectral energy* in the spatial-frequency domain. Based on this observation, we propose **CEN** (Algorithm 1), a frequency-domain autofocus method designed to extract, stabilize, and localize this migration cue under sensor noise, sparsity, scene dynamics, and illumination drift.

## 3.1. Focus-induced spectral migration in spike measurements

A spike camera outputs binary measurements $S_t \in \{0,1\}^{H \times W}$ at each sampling instant $t$ (Dong et al., 2019). To obtain a stable representation for analysis, we partition a focus sweep into $B$ non-overlapping blocks, each aggregating $dt$ consecutive spike frames, and compute one scalar focus score for each block. The accumulated spike block $b$ is defined as

$$A_b = \sum_{t=b \cdot dt}^{(b+1) \cdot dt - 1} S_t, \qquad b = 0, \dots, B-1. \qquad (1)$$

Spike-based autofocus is posed as identifying the block index $\hat{b}$ corresponding to the best-focused state during the focus sweep. We provide a simple optical interpretation for why spectral-centroid migration is expected to occur near focus. Under a standard defocus model, the observed signal can be written as the convolution between a latent sharp image $I$ and a defocus point spread function $h_\sigma$:

$$I_\sigma(x) = (I * h_\sigma)(x), \qquad (2)$$

where $\sigma$ denotes the blur scale. In the frequency domain, the corresponding power spectrum is

$$P_\sigma(\omega) = |\hat{I}(\omega)|^2 |\hat{h}_\sigma(\omega)|^2. \qquad (3)$$

For typical defocus blur, $|\hat{h}_\sigma(\omega)|^2$ behaves as a low-pass attenuation term. Using a Gaussian approximation, this can be written as

$$P_\sigma(\omega) = |\hat{I}(\omega)|^2 \exp(-\sigma^2 \|\omega\|^2), \qquad (4)$$

showing that high-frequency components are increasingly suppressed as $\sigma$ grows. This motivates measuring the frequency center of mass within a bounded region:

$$C_r(\sigma) = \frac{\int_{\|\omega\| \le r} \|\omega\| P_\sigma(\omega) \, d\omega}{\int_{\|\omega\| \le r} P_\sigma(\omega) \, d\omega + \varepsilon}. \qquad (5)$$

---

**Algorithm 1** CEN: Autofocus via Structure-Consistent Spectral Migration

---

**Require:** Spike sequence $\{S_t\}$, block size $dt$, candidate radii $\mathcal{R}$
**Ensure:** Predicted focus block index $\hat{b}$
1: Partition $\{S_t\}$ into blocks and form accumulated spike blocks $\{A_b\}$ (Eq. 1)
2: **for** $b = 0$ to $B-1$ **do**
3:     Remove blockwise mean from $A_b$ (Eq. 6)
4:     Compute FFT-shifted power spectrum $P_b$ (Eq. 7)
5:     Compute spectral-centroid responses $\{\text{CEN}_r(b)\}_{r \in \mathcal{R}}$ (Eq. 9)
6: **end for**
7: Normalize each response curve $\{\text{CEN}_r(b)\}_b$ to obtain $y_r$ (Eq. 10)
8: Evaluate structure-consistency score $\mathcal{S}(y_r)$ for each $r \in \mathcal{R}$ (Eq. 11)
9: Select the most consistent response $r^\star$ (Eq. 12)
10: Identify near-maximum region $\Omega$ on $y_{r^\star}$ (Eq. 13)
11: Localize focus by weighted centroid aggregation to obtain $\hat{b}$ (Eq. 14)
12: **Return** predicted focus block $\hat{b}$

---

During a focus sweep, $\sigma$ decreases when approaching focus and increases after passing the focal plane. Therefore, spectral energy first migrates outward and then inward, and the centroid exhibits an increase-then-decrease trend with a peak near the best-focused state. Although spike measurements are quantized, sparse, and affected by noise or motion, this focus-induced spectral migration remains statistically observable in accumulated spike blocks, which forms the foundation of our autofocus formulation.

## 3.2. Spectral centroid as a migration statistic

To quantify spectral migration, we analyze the power spectrum of each block. Let $\mathcal{F}(\cdot)$ denote the 2D discrete Fourier transform (Bracewell, 1989; Cochran et al., 1967). To suppress global bias caused by illumination changes, we first remove the blockwise DC (direct-current) component by subtracting the mean:

$$\tilde{A}_b = A_b - \mu_b, \qquad (6)$$

where $\mu_b = \frac{1}{HW} \sum_{i,j} (A_b)_{i,j}$ denotes the spatial mean of block $b$.

We then compute the centered power spectrum by applying an FFT shift, which places the zero-frequency (DC) component at the spectrum center and enables radially symmetric frequency analysis:

$$P_b(u,v) = \left| \text{FFT}_{\text{shift}}(\mathcal{F}(\tilde{A}_b))(u,v) \right|^2. \qquad (7)$$

Let $\rho(u,v)$ denote the normalized radial spatial frequency,

$$\rho(u,v) \;=\; \sqrt{\left(\frac{u-u_0}{W}\right)^2 + \left(\frac{v-v_0}{H}\right)^2}, \qquad (8)$$

where $(u_0, v_0)$ is the spectrum center after FFT shift. We summarize the spectral distribution by its centroid within a bounded frequency region:

$$\mathrm{CEN}_r(b) = \frac{\sum_{(u,v):\,\rho(u,v)\leq r} \rho(u,v)\, P_b(u,v)}{\sum_{(u,v):\,\rho(u,v)\leq r} P_b(u,v) + \varepsilon}. \qquad (9)$$

Here $r$ denotes a spectral radius that upper-bounds the normalized radial frequency ($\rho(u,v) \leq r$), controlling the frequency range over which the centroid is computed; its selection is addressed in Sec. 3.3. The constant $\varepsilon$ is a small positive value prevented division by zero when spectral energy within the bounded region is weak. This statistic captures the dominant frequency scale of the spike signal. As focus improves, energy shifts outward and $\mathrm{CEN}_r(b)$ increases; as defocus increases, the centroid shifts inward. Importantly, this formulation avoids image reconstruction and does not rely on spatial gradients, making it naturally compatible with sparse binary measurements.

### 3.3. Structure-consistent identification of the focus cue

In an ideal scenario, the centroid response over blocks would exhibit a single, well-localized extremum at the true focus position. In practice, spike measurements often violate this behavior. Sensor noise, motion, and illumination drift can introduce spurious components and long-range bias, leading to response curves with multiple extrema, broad plateaus, or boundary-dominated trends. As a result, the central challenge is not merely to compute a spectral statistic, but to identify the response that most faithfully reflects focus-induced spectral-centroid migration.

**Principles.** We evaluate centroid responses under a small set of admissible frequency bounds and select the one whose shape is most consistent with the expected migration pattern. Rather than assuming an idealized unimodal profile, we characterize a reliable focus response by: (i) **Distinctiveness**: a dominant extremum stands out from the typical interior level; (ii) **Localization**: the extremum is spatially concentrated, which is quantified by penalties on both peak width and near-maximum plateau behavior; (iii) **Sharpness**: curvature around the extremum is pronounced; (iv) **Interior consistency**: the extremum does not arise from sweep boundaries.

**Candidate set and normalization.** Let $\mathcal{R}$ denote a set of admissible frequency bounds, and let $y_r \in \mathbb{R}^B$ be the normalized spectral-centroid response

$$y_r(b) = \frac{\mathrm{CEN}_r(b) - \min_b \mathrm{CEN}_r(b)}{\max_b \mathrm{CEN}_r(b) - \min_b \mathrm{CEN}_r(b) + \varepsilon}. \qquad (10)$$

**Structure-consistency score.** We define a structure-consistency functional as a weighted combination of five terms,

$$\mathcal{S}(y_r) = \alpha_{\mathrm{prom}}\, \mathcal{S}_{\mathrm{prom}}(y_r) + \alpha_{\mathrm{curv}}\, \mathcal{S}_{\mathrm{curv}}(y_r) - \alpha_{\mathrm{width}}$$
$$\mathcal{S}_{\mathrm{width}}(y_r) - \alpha_{\mathrm{plat}}\, \mathcal{S}_{\mathrm{plat}}(y_r) - \alpha_{\mathrm{edge}}\, \mathcal{S}_{\mathrm{edge}}(y_r). \qquad (11)$$

where: $\mathcal{S}_{\mathrm{prom}}$ measures peak prominence relative to the interior baseline, $\mathcal{S}_{\mathrm{curv}}$ favors a spatially concentrated extremum via local curvature, $\mathcal{S}_{\mathrm{width}}$ penalizes overly narrow or overly broad peaks using a width-to-length ratio, $\mathcal{S}_{\mathrm{plat}}$ penalizes plateau-like near-maximum regions, and $\mathcal{S}_{\mathrm{edge}}$ penalizes extrema near sweep boundaries. All terms are computed directly from the one-dimensional response shape and are scene-independent. We use fixed weights (($\alpha_{\mathrm{prom}}, \alpha_{\mathrm{curv}}, \alpha_{\mathrm{width}}, \alpha_{\mathrm{plat}}, \alpha_{\mathrm{edge}}$) = (1.10, 0.35, 0.45, 0.35, 0.60)) across all scenes to avoid per-sequence tuning.

**Selecting the structurally consistent response.** The final response is selected by maximizing structure consistency:

$$r^\star = \arg\max_{r \in \mathcal{R}} \mathcal{S}(y_r), \qquad (12)$$

and the corresponding $y_{r^\star}$ is used for focus localization. This selection provides an unsupervised strategy for adapting the frequency radius to each sequence, whose necessity is verified in ablation studies. This selection explicitly favors responses that best reflect focus-induced spectral-centroid migration, without requiring an unimodal response.

### 3.4. Robust focus localization

Even after identifying a structure-consistent response, the peak region may span multiple adjacent blocks due to noise or gradual spectral variation. To avoid brittle single-point decisions, we localize the focus by aggregating the near-maximum region.

Let $p_0 = \arg\max_b y_{r^\star}(b)$ and define

$$\Omega = \left\{ b \in \{0, \ldots, B-1\} \;:\; y_{r^\star}(b) \geq (1-\tau)\, y_{r^\star}(p_0) \right\}, \qquad (13)$$

where $\tau$ is a small tolerance. The predicted focus block is computed as a weighted centroid

$$\hat{b} = \mathrm{round}\left(\frac{\sum_{b \in \Omega} b\, w_b}{\sum_{b \in \Omega} w_b + \varepsilon}\right), \qquad w_b = \left(y_{r^\star}(b)\right)^\gamma. \qquad (14)$$

This estimator stabilizes localization under flat or irregular peaks and yields robust predictions across long sweeps and dynamic scenes (illustration in Appendix A.2).

# 4. Experiment

## 4.1. Main Comparison on Autofocus Accuracy

**Datasets.** We evaluate autofocus on both synthetic and real spike-camera focus sweeps.

- **Synthetic.** We use focus-sweep sequences from the CVIA dataset (Pertuz et al., 2013a). Each sequence contains 25–50 images captured at different focus distances, with the focus distance of each image recorded in meters. We convert each image into a spike sequence using a noise-aware integrate-and-fire simulation (Liu et al., 2025). We simulate moderate- and low-light settings by varying the light factor, where a smaller light factor corresponds to a lower photon rate and produces sparser spike responses.

- **Real.** We use spike-camera sequences from the SAD dataset (Su et al., 2024), which spans static and dynamic scenes with diverse illumination variations. Block-level ground-truth focal positions enable quantitative evaluation.

**Compared methods.** We compare the proposed method (**CEN**) with representative autofocus criteria operating in different domains:

- **Spatial-domain**: **GRAD** (Tenengrad) and **LAP** (Laplacian energy), which measure spatial sharpness on accumulated block image.

- **Temporal-domain**: **SD** (Spike Dispersion) (Su et al., 2024), defined as the variance of accumulated spike counts normalized by the squared mean intensity. **COUNT** (spike count), which exploit temporal statistics of spike events during focus sweeping.

- **Frequency-domain**: **MFDCT** (multi-frequency DCT energy) and **HF** (high-frequency energy), which estimate focus quality based on spectral responses.

In addition to these direct spike-domain or accumulated-block baselines, we further evaluate reconstruction-based image autofocus pipelines on real spike-camera data. Since conventional image-domain focus measures require intensity images as input, we first reconstruct an image from each accumulated spike block using **TFP** (Zhu et al., 2023) or **TFI** (Zhu et al., 2019), and then apply representative image-based focus measures, including **GRAD** (Tenengrad), **LAP** (Laplacian), **RDF** (Jeon et al., 2020), and **DRDF** (Ashfaq & Tariq Mahmood, 2026).

**Implementation details.** SD (Su et al., 2024), RDF (Jeon et al., 2020), and DRDF (Ashfaq & Tariq Mahmood, 2026) are evaluated using the authors' officially released implementations. All other methods are independently implemented following their original formulations and descriptions. All focus-response curves are constructed at the block

*Table 1.* Focus estimation error on synthetic focus sweeps under moderate-light and low-light conditions. Error is reported in centimeters; lower is better.

| Method | Moderate-light | | | | Low-light | | | |
|---|---|---|---|---|---|---|---|---|
| | simu01 | simu02 | simu03 | **Mean** | simu01 | simu02 | simu03 | **Mean** |
| GRAD | 0.37 | 0.00 | 0.63 | 0.33 | 4.00 | 4.00 | 6.30 | 4.80 |
| HF | 0.37 | 0.37 | 0.00 | 0.24 | **0.37** | 0.37 | 5.00 | 1.90 |
| LAP | 0.37 | 2.20 | 0.00 | 0.86 | 4.00 | 4.00 | 5.60 | 4.60 |
| MFDCT | 0.37 | 0.37 | 0.00 | 0.24 | 4.00 | 4.00 | 5.60 | 4.60 |
| SD | 0.37 | 0.37 | 0.00 | 0.24 | 12.00 | 2.20 | 6.30 | 6.90 |
| COUNT | 4.00 | 4.00 | 0.63 | 2.90 | 12.00 | 6.60 | 2.50 | 7.20 |
| CEN | **0.00** | **0.00** | **0.00** | **0.00** | 2.90 | **0.37** | **0.63** | **1.30** |

level, where each method produces one scalar focus score per accumulated block. For spatial-response methods, the scalar score is obtained by aggregating the response map over all pixels; for temporal- and frequency-domain methods, it is computed directly from block-level statistics or spectra. To ensure a fair comparison, all methods are evaluated under identical block partitioning $dt$, following the configuration provided by the SAD dataset. The main CEN parameters, including the candidate frequency radii and structure-consistency weights, are kept fixed across synthetic and real experiments. Dataset-specific settings are limited to the block partitioning protocol.

## 4.2. Evaluation on Synthetic Focus Sweeps

We first evaluate CEN on synthetic focus sweeps. We use three synthetic sequences and evaluate two illumination settings, moderate-light and low-light. Table 1 reports the focus estimation error on synthetic focus sweeps. Under moderate-light conditions, most baselines produce reasonable responses because structural information is relatively well preserved, while CEN achieves zero error on all sequences. Under low-light conditions, spike signals become sparse and noisy, making local-gradient and high-frequency responses less stable. In contrast, CEN obtains the lowest mean error, indicating that the spectral-centroid migration cue remains more stable under degraded illumination.

**Quantitative comparison.** Table 2 reports autofocus accuracy across representative real spike-camera scenes. Overall, CEN achieves the lowest mean error among all compared direct baselines. We note that the improvement is not uniform in every individual scene. For example, HF performs slightly better on USAF, where regular high-frequency patterns make fixed-band spectral energy effective, and SD remains competitive in several controlled settings. However, these baselines become less reliable when their assumptions are violated: HF is sensitive to burst-like or noise-induced high-frequency responses, while SD can degrade under motion or complex textures. In contrast, CEN captures the global spectral-centroid migration pattern over the focus sweep, leading to more stable average performance across diverse scenes. Detailed per-sequence results are provided in Appendix A.1.

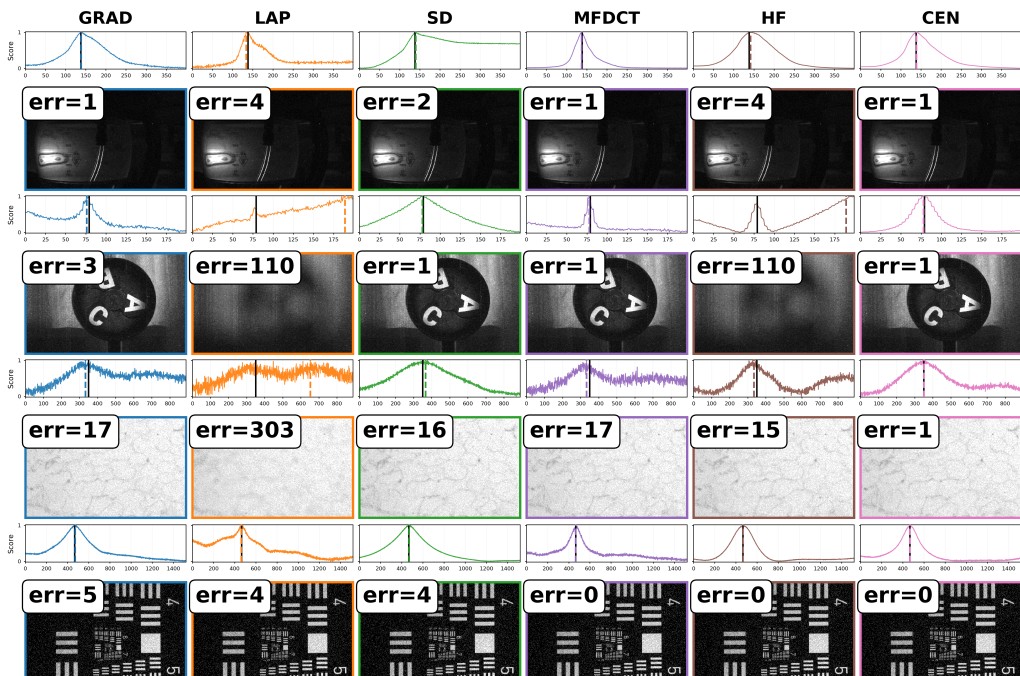

*Figure 2.* Qualitative comparison of autofocus responses and predicted focus blocks on representative spike-camera scenes. Each column corresponds to one method. For each scene (Bottle, Fan, Lily, USAF), the top subplot shows the normalized response curve over blocks with the ground-truth focus block indicated by a black vertical line and the method prediction indicated by a colored dashed line. The bottom subplot visualizes the accumulated block image at the predicted focus, and the block-level absolute error $|\hat{b} - b^*|$ is annotated in the corner. Overall, CEN produces a sharper and better-aligned extremum and yields more accurate focus blocks across diverse scenes.

*Table 2.* Autofocus accuracy across representative spike-camera scenes. Mean AbsErr (RelErr, %) is reported; lower is better. Best and second-best results are shown in bold and underlined.

| Scene | GRAD | LAP | SD | COUNT | MFDCT | HF | CEN |
|---|---|---|---|---|---|---|---|
| Bottle | **1.00 (0.3)** | 2.00 (0.5) | **1.00 (0.2)** | 129.50 (32.5) | **1.00 (0.3)** | 3.50 (0.9) | **1.00 (0.3)** |
| Lily | 20.50 (2.3) | 298.00 (33.6) | 12.00 (1.4) | 427.50 (48.0) | 18.00 (2.0) | 10.50 (1.2) | **2.50 (0.3)** |
| Fan | 4.50 (2.2) | 88.50 (44.5) | 2.50 (1.2) | 79.50 (40.0) | 3.50 (1.8) | 88.50 (44.5) | **2.00 (1.0)** |
| USAF | 368.67 (24.3) | 386.11 (25.4) | 7.67 (0.5) | 729.56 (48.0) | 500.89 (33.0) | **3.33 (0.2)** | 3.44 (0.2) |
| MEAN | 224.67 (15.2) | 283.47 (25.7) | 6.67 (0.7) | 522.60 (44.9) | 303.53 (20.3) | 15.67 (6.3) | **2.80 (0.3)** |

*Table 3.* Additional comparison with reconstruction-based image autofocus baselines. Mean AbsErr (RelErr, %) is reported; lower is better. The left block uses direct count-normalized reconstruction, while the right block uses TFI reconstruction. Best and second-best results in each row are shown in bold and underlined, respectively.

| Scene | Count-normalized reconstruction | | | | TFI reconstruction | | | | CEN |
|---|---|---|---|---|---|---|---|---|---|
| | GRAD | LAP | RDF | DRDF | GRAD | LAP | RDF | DRDF | |
| Bottle | **0.50 (0.3)** | 1.50 (1.0) | 2.00 (1.4) | 2.00 (1.4) | 1.00 (0.7) | 1.00 (0.7) | 4.50 (3.0) | 3.50 (2.3) | 1.00 (0.7) |
| Lily | 13.50 (3.8) | 330.50 (92.7) | 14.50 (4.1) | 17.00 (4.7) | 34.50 (9.6) | 340.00 (95.0) | 150.00 (42.8) | 11.50 (3.3) | **2.50 (0.7)** |
| Fan | 3.50 (3.1) | 88.50 (89.5) | 4.50 (4.4) | 52.00 (43.3) | 89.50 (87.0) | 89.50 (87.0) | 91.00 (92.0) | 91.00 (92.0) | **2.00 (1.9)** |
| USAF | 43.11 (6.7) | 550.11 (91.6) | 308.78 (50.2) | 447.11 (70.2) | 95.22 (14.1) | 138.78 (19.6) | 85.56 (12.7) | 85.56 (12.7) | **3.44 (0.5)** |
| MEAN | 28.20 (5.0) | 386.13 (79.4) | 188.07 (31.4) | 277.73 (48.7) | 73.80 (21.4) | 140.67 (36.1) | 84.07 (26.0) | 65.47 (20.7) | **2.80 (0.8)** |

We further compare CEN with reconstruction-based image autofocus pipelines. Table 3 summarizes the grouped results. Overall, CEN achieves the lowest mean error among all com-

pared reconstruction-based pipelines. Although RDF and DRDF are designed to improve robustness to noise and local structural variations in conventional image-based autofocus,

*Table 4.* Focus localization quality measured by GT ratio (higher is better) and GT rank (lower is better), reported as *Ratio / Rank*.

| Scene | GRAD | LAP | SD | COUNT | MFDCT | HF | CEN |
|---|---|---|---|---|---|---|---|
| Bottle | 0.982 / 4.50 | 0.973 / 5.00 | **0.992 / 2.00** | 0.456 / 170.00 | 0.972 / 3.00 | 0.975 / 10.00 | 0.986 / 6.00 |
| Lily | 0.835 / 66.50 | 0.665 / 297.50 | **0.936 / 50.00** | 0.425 / 533.00 | 0.779 / 56.50 | 0.896 / **34.50** | 0.923 / 37.50 |
| Fan | 0.768 / 39.00 | 0.646 / 67.00 | 0.979 / 5.00 | 0.292 / 100.50 | 0.852 / 5.00 | 0.681 / 45.50 | **0.996 / 2.50** |
| USAF | 0.622 / 470.00 | 0.610 / 538.44 | 0.985 / **15.78** | 0.166 / 1174.67 | 0.346 / 627.33 | 0.978 / **15.78** | **0.987** / 16.00 |
| MEAN | 0.717 / 296.67 | 0.670 / 372.33 | 0.979 / 17.07 | 0.256 / 811.93 | 0.642 / 385.00 | 0.927 / 21.47 | **0.980 / 15.73** |

*Table 5.* Autofocus accuracy (AbsErr (RelErr) ↓) under static and dynamic scenes.

| Motion Type | GRAD | LAP | SD | COUNT | MFDCT | HF | CEN |
|---|---|---|---|---|---|---|---|
| Static | 391.00 (23.26) | 183.71 (30.43) | 3.00 (0.39) | 541.71 (46.34) | 356.86 (32.30) | 33.86 (15.46) | **3.14 (0.33)** |
| Dynamic | 201.63 (13.70) | 245.13 (21.61) | 9.50 (0.94) | 505.88 (43.72) | 256.88 (16.25) | 12.25 (4.62) | **2.50 (0.36)** |

their performance here is constrained by the intermediate spike-to-image reconstruction. In particular, reconstruction may introduce smoothing artifacts and temporal aggregation effects, which can distort fine-scale spatial structures that image-domain focus measures rely on. By contrast, CEN directly operates on spike measurements and exploits the sensor-observable spectral-migration cue, thereby avoiding reconstruction-induced information loss and yielding more stable focus estimation.

**Qualitative analysis.** Figure 2 visualizes response curves and predicted focus blocks for representative scenes. Across scenes, several comparisons exhibit unstable responses, which can shift the selected focus away from the ground truth and lead to visibly less focused accumulated blocks. For example, in dynamic scenes such as Fan, burst-like spike activity can introduce abnormal high-frequency responses, causing fixed-band HF measures to select spurious maxima. In contrast, CEN produces a more stable response whose dominant peak is consistently aligned with the ground-truth block, yielding small errors across all shown scenes (err=1 or 0) and clearer focused outputs. Notably, when multiple methods perform well (e.g., the USAF example), CEN remains competitive and achieves the correct focus block.

### 4.3. Response stability and Robustness.

Table 4 evaluates response stability and discriminability. The GT ratio measures how prominently the ground-truth focus block stands out, while the GT rank reflects its relative ordering among all candidates. CEN achieves the highest average GT ratio and the lowest average GT rank, indicating that the ground-truth focus position is consistently emphasized as a clear and dominant extremum.

We further evaluate robustness under motion and illumination variations. As shown in Table 5 and Figure 3, CEN achieves the lowest error on both static and dynamic scenes

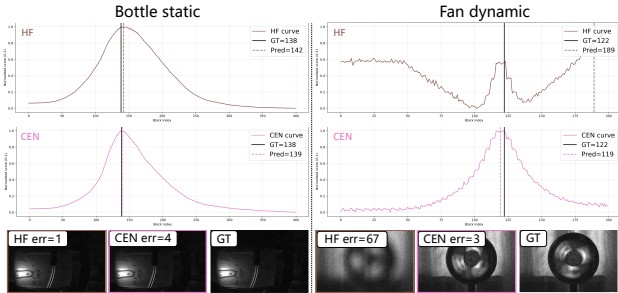

*Figure 3.* Static vs. dynamic qualitative comparison (Bottle vs. Fan). We visualize the focus-response curves of HF and CEN, with GT (solid line) and prediction (dashed line). Bottom: predicted-focus spike images and GT; absolute block error is shown in the upper-left.

and consistently predicts focus blocks closer to the ground truth. Robustness to illumination variation is summarized in Table 6 and Figure 4, where CEN remains accurate across all illumination patterns, while several baselines exhibit biased or unstable responses.

Motion blur and defocus blur may both attenuate high-frequency components at a single observation. However, CEN does not rely on a single spectral snapshot. Defocus is coupled with lens position and induces a structured spectral-centroid migration over focus sweep, whereas motion-induced blur is generally not sweep-aligned. Dynamic-scene results suggest that CEN can still identify the focus-induced trajectory when such structured evolution is present.

### 4.4. Computational efficiency and complexity.

We evaluate the computational efficiency of different autofocus criteria under identical experimental settings. Runtime is reported in milliseconds per block on an Apple M2 Mac-Book Air with 8 GB unified memory. Spatial- and temporal-domain methods are the most efficient: GRAD and LAP

*Table 6.* Autofocus accuracy (AbsErr (RelErr) ↓) under different illumination variation patterns during focus sweeping.

| Illumination | GRAD | LAP | SD | COUNT | MFDCT | HF | CEN |
|---|---|---|---|---|---|---|---|
| Constant | 98.38 (7.55) | 188.38 (29.32) | 4.38 (0.79) | 327.00 (41.23) | 96.88 (7.03) | 25.75 (11.66) | **1.50 (0.40)** |
| Decrease | 242.00 (15.90) | 743.00 (48.90) | **6.50 (0.40)** | 624.00 (41.10) | 620.00 (40.80) | 7.00 (0.45) | 7.50 (0.50) |
| Increase | 21.50 (1.40) | 403.00 (26.55) | **1.50 (0.10)** | 881.00 (58.00) | 239.00 (15.75) | 3.00 (0.20) | **1.50 (0.10)** |
| Minor | 753.50 (49.60) | 147.50 (9.70) | 4.00 (0.25) | 754.50 (49.70) | 758.50 (49.95) | **0.50 (0.05)** | 2.50 (0.20) |
| Significant | 549.00 (36.10) | 154.00 (10.10) | 37.00 (2.40) | 696.00 (45.80) | 543.00 (35.70) | 8.00 (0.50) | **7.00 (0.50)** |

take 11.42 ms and 11.59 ms per block on average, respectively, while SD and COUNT are slightly faster at 10.49 ms and 9.90 ms. These methods involve only block-wise accumulation or local filtering and therefore exhibit linear complexity with respect to the number of pixels ($O(N)$).

Spectral-domain methods require one Fourier transform per block and therefore scale as $O(N \log N)$. MFDCT and HF run at 15.40 ms and 18.44 ms per block, respectively, while the proposed CEN takes 20.53 ms. The additional cost mainly comes from the block-wise 2D FFT, radial aggregation, and response-curve analysis.

Overall, CEN trades additional spectral computation for improved autofocus accuracy and robustness. This trade-off should be considered when deploying the method on resource-constrained neuromorphic edge systems.

### 4.5. Ablation Studies

We conduct ablation experiments to verify the necessity of key design components in CEN, including (i) DC removal and spectrum centering for illumination robustness, (ii) the selection of the spectral radius $r$, and (iii) structural terms used to identify a focus-discriminative response curve. All results follow the same block partitioning and evaluation protocol as in Sec. 4.1.

**Ablation on DC removal and FFT shift.** Spike sequences in SAD exhibit diverse illumination variations, which can introduce global bias and low-frequency dominance in accumulated spike blocks. We ablate two implementation choices affecting the stability of the frequency-domain statistic. *DC removal* suppresses global bias by subtracting the block-wise mean, while *FFT shift* centers the DC component to ensure consistent radial frequency definition. Without FFT shift, radial aggregation becomes misaligned with the true frequency geometry, leading to distorted centroid responses. Using DC removal alone yields a mean absolute error of 35.47 blocks, while FFT shift alone reduces the error to 6.67 blocks but remains sensitive to DC bias. Combining both achieves the lowest error (2.80 blocks), demonstrating that proper DC handling and spectrum centering are critical for stable autofocus under illumination variation.

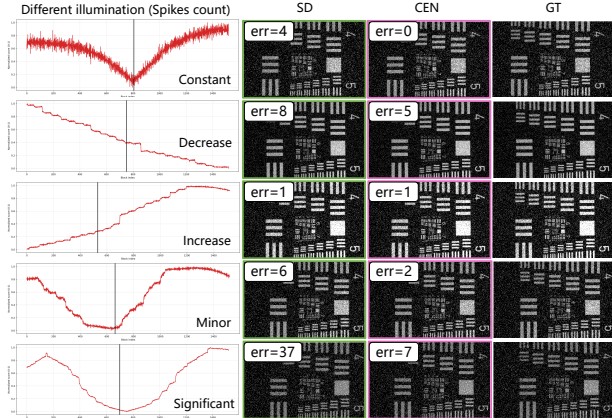

*Figure 4.* Autofocus comparison under different illumination changes. Each row shows one illumination pattern (constant, decrease, increase, minor, significant). From left to right: spike count curve, SD result, CEN result, and GT. Absolute block error is shown in the upper-left of each estimated focus image.

**Ablation on spectral radius $r$.** CEN computes the spectral centroid within a bounded frequency radius $r$. A large radius tends to absorb high-frequency noise, while a small radius may discard focus-relevant structures. We therefore select $r$ automatically using a unsupervised curve-shape criterion. Figure 5 compares automatic radius selection, a fixed radius ($r = 0.25$), and an unconstrained setting. Auto-$r$ consistently reduces error across scenes and significantly outperforms the fixed setting. The comparison with the unconstrained case further confirms that radius control is necessary to prevent high-frequency residuals from misleading focus localization.

To provide intuition, Figure 6 illustrates the effect of radius truncation in the frequency domain. The retained component within $r^\star$ preserves focus-relevant structures, while the residual outside ($r > r^\star$) is dominated by noise-like high-frequency patterns. This comparison highlights the role of proper radius selection in suppressing distracting spectral components for reliable focus estimation. Importantly, CEN does not rely on the magnitude of high-frequency energy at a single focus step. Instead, it uses the structured migration pattern of the bounded spectral centroid across the focus sweep, which makes it less sensitive to incidental texture or

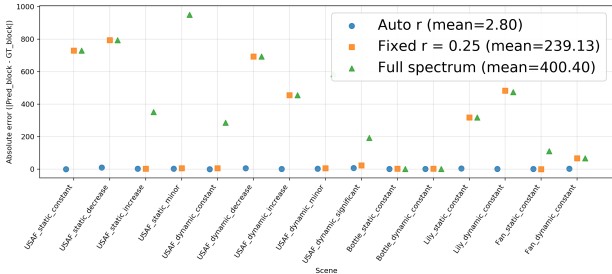

*Figure 5.* Ablation on the spectral radius $r$. We compare three settings: automatic radius selection (Auto-$r$), a fixed radius ($r = 0.25$), and using the full spectrum. Each marker corresponds to one scene, and the y-axis reports the absolute block error. The legend shows the mean error over all scenes.

noise-induced high-frequency fluctuations.

**Ablation on curve-structure criteria.** CEN selects a focus-discriminative response curve using structural criteria that favor a clear dominant extremum and penalize ambiguous shapes such as wide peaks, plateaus, or edge-biased extrema. Table 7 ablates each structural term.

Using only peak prominence (with or without width) is insufficient and leads to large errors due to multiple peaks or broad plateaus. Removing any individual term degrades performance, while the full design achieves the lowest mean error. These results validate the necessity of each structural component for reliable curve selection.

### 4.6. Limitations.

**Multi-depth and motion-dominated scenes.** Multi-depth scenes and complex motion may reduce the distinctiveness of the spectral-centroid response. Multi-depth scenes can produce a superposition of responses from different depth layers, making the focus cue flatter when several depths contribute comparably. Strong irregular motion can further introduce frequency components that do not follow the consistent migration pattern induced by focus change. Although the structure-consistency criterion helps favor stable and localized spectral evolution, extreme multi-depth or motion-dominated cases remain challenging and will be explored in future work.

**Edge deployment.** The runtime analysis shows that CEN is practical for block-wise CPU processing and remains comparable to other spectral baselines. However, compared with lightweight $O(N)$ temporal metrics such as SD or COUNT, CEN relies on repeated block-wise 2D FFTs and therefore has a less favorable hardware profile in terms of compute and memory access. Its latency and energy cost under strict microsecond- or microjoule-level edge constraints remain to be verified in future work.

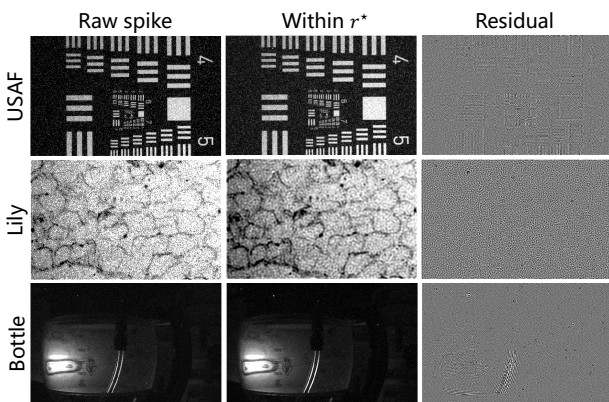

*Figure 6.* Effect of radius truncation in the frequency domain. Rows show representative scenes. From left to right: accumulated spike blocks, band-limited component within $r^\star$, and high-frequency residual outside $r^\star$ dominated by noise-like patterns.

*Table 7.* Ablation of structural criteria used for focus curve selection.

| Variant | prom | width | plateau | edge_pos | Err $\downarrow$ |
|---|---|---|---|---|---|
| prom only | ✓ | | | | 56.47 |
| prom+width | ✓ | ✓ | | | 33.40 |
| Full - prom | | ✓ | ✓ | ✓ | 35.27 |
| Full - edge_pos | ✓ | ✓ | ✓ | | 33.07 |
| Full - plateau | ✓ | ✓ | | ✓ | 3.13 |
| Full - width | ✓ | | ✓ | ✓ | 3.07 |
| Full | ✓ | ✓ | ✓ | ✓ | 2.80 |

## 5. Conclusion

We studied autofocus for spike cameras from the perspective of what is reliably observable from sparse binary measurements during a focus sweep. We showed that a stable and physically meaningful cue is the focus-induced migration of spectral energy in the frequency domain, which can be compactly summarized by a bounded spectral centroid. Building on this observation, we proposed **CEN**, which measures spectral-centroid migration on accumulated spike blocks without image reconstruction or explicit edge extraction. To handle multi-peak and irregular responses in real scenes, CEN further performs structure-consistent response identification to select a focus-discriminative frequency bound and localizes the focus using a robust weighted near-maximum centroid. Experiments on both synthetic and real spike-camera datasets demonstrate that CEN achieves the best overall autofocus accuracy and improves response discriminability across diverse scenes, motion types, and illumination variations. Future work will extend the approach to adaptive block sizing and online focus control, and explore joint designs with depth and motion estimation for broader neuromorphic perception pipelines.

## Impact Statement

This work advances autofocus for spike cameras by improving focus estimation directly from sparse spike measurements, without image reconstruction. The proposed method may benefit low-power and high-speed vision systems, including robotics, intelligent sensing, and neuromorphic imaging, where reliable focus control is important under motion and illumination changes. We do not foresee direct negative societal impacts beyond the general considerations associated with visual sensing technologies. As with other imaging systems, potential deployment should respect privacy, safety, and responsible data collection practices.

## Acknowledgments

This work was supported by the Shenzhen Science and Technology Program (KQTD 20240729102051063), the National Natural Science Foundation of China (No. 62332002, No. 62425101) and the China Scholarship Council.

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

# A. Appendix.

## A.1. Quantitative Evaluation of Autofocus Accuracy and Focus Localization

*Table 8.* Autofocus error: AbsErr (RelErr). Lower is better. Best AbsErr per row is bold; second best is underlined.

| Scene | GRAD | LAP | SD | COUNT | MFDCT | HF | CEN |
|---|---|---|---|---|---|---|---|
| USAF_static_constant | 730 (48.1) | 730 (48.1) | 4 (0.3) | 606 (39.9) | 730 (48.1) | **1 (0.1)** | **1 (0.1)** |
| USAF_static_decrease | 37 (2.4) | 793 (52.2) | **5 (0.3)** | 574 (37.8) | 793 (52.2) | 9 (0.6) | 10 (0.7) |
| USAF_static_increase | 40 (2.6) | 351 (23.1) | **2 (0.1)** | 1016 (66.9) | 23 (1.5) | 6 (0.4) | **2 (0.1)** |
| USAF_static_minor | 922 (60.7) | 5 (0.3) | 2 (0.1) | 926 (61.0) | 932 (61.4) | **1 (0.1)** | 3 (0.2) |
| USAF_dynamic_constant | 5 (0.3) | 4 (0.3) | 4 (0.3) | 745 (49.0) | **0 (0)** | **0 (0)** | **0 (0)** |
| USAF_dynamic_decrease | 447 (29.4) | 693 (45.6) | 8 (0.5) | 674 (44.4) | 447 (29.4) | **5 (0.3)** | **5 (0.3)** |
| USAF_dynamic_increase | 3 (0.2) | 455 (30.0) | 1 (0.1) | 746 (49.1) | 455 (30.0) | **0 (0)** | 1 (0.1) |
| USAF_dynamic_minor | 585 (38.5) | 290 (19.1) | 6 (0.4) | 583 (38.4) | 585 (38.5) | **0 (0)** | 2 (0.1) |
| USAF_dynamic_significant | 549 (36.1) | 154 (10.1) | 37 (2.4) | 696 (45.8) | 543 (35.7) | 8 (0.5) | **7 (0.5)** |
| Bottle_static_constant | **1 (0.3)** | 4 (1.0) | 2 (0.5) | 118 (29.6) | **1 (0.3)** | 4 (1.0) | **1 (0.3)** |
| Bottle_dynamic_constant | 1 (0.3) | **0 (0)** | **0 (0)** | 141 (35.3) | 1 (0.3) | 3 (0.8) | 1 (0.3) |
| Lily_static_constant | 24 (2.7) | 293 (33.0) | 8 (0.9) | 483 (54.3) | 19 (2.1) | 6 (0.7) | **4 (0.4)** |
| Lily_dynamic_constant | 17 (1.9) | 303 (34.1) | 16 (1.8) | 372 (41.8) | 17 (1.9) | 15 (1.7) | **1 (0.1)** |
| Fan_static_constant | 3 (1.5) | 110 (55.3) | **1 (0.5)** | 69 (34.7) | **1 (0.5)** | 110 (55.3) | **1 (0.5)** |
| Fan_dynamic_constant | 6 (3.0) | 67 (33.7) | 4 (2.0) | 90 (45.2) | 6 (3.0) | 67 (33.7) | **3 (1.5)** |
| MEAN | 224.67 (15.2) | 283.47 (25.7) | 6.67 (0.7) | 522.6 (44.9) | 303.53 (20.3) | 15.67 (6.3) | **2.8 (0.3)** |

Tables 8 and 9 report the quantitative comparison of seven classical focus measures (GRAD, LAP, SD, COUNT, MFDCT, HF, and CEN) on scenes under diverse static and dynamic conditions.

Table 8 evaluates autofocus accuracy using absolute error (AbsErr) and relative error (RelErr). Lower values indicate more precise focus estimation. The results show that CEN consistently achieves the lowest mean error (2.8 / 0.3), outperforming all competing methods by a clear margin, while SD ranks second. Traditional gradient- or frequency-based measures (e.g., GRAD, LAP, COUNT, MFDCT) exhibit large errors, particularly in dynamic or low-contrast scenarios, indicating limited robustness to defocus variation and scene motion.

Table 9 further assesses focus localization quality using the ground-truth ratio (GT ratio) and ground-truth rank (GT rank). Higher ratios and lower ranks correspond to more accurate and reliable localization of the true focal position. CEN again achieves the best overall performance with the highest mean GT ratio (0.980) and the lowest mean rank (15.73), followed closely by SD. In contrast, COUNT and several classical operators yield poor localization accuracy, often ranking far from the ground truth.

Overall, the two tables jointly demonstrate that CEN provides both superior numerical accuracy and more reliable focus localization across a wide range of imaging conditions, validating its robustness and suitability for precise autofocus in practical dynamic scenes.

## A.2. Robust localization analysis.

Figure 7 illustrates the proposed robust focus localization strategy based on near-maximum aggregation. Instead of relying on a single argmax point, which can be unstable under noise, discretization, or gradual spectral variation, we identify a near-maximum region $\Omega$ around the response peak and estimate the focus position as a weighted centroid over this region.

As shown in the figure, the raw CEN response exhibits a broad and slightly asymmetric peak. The argmax (gray dashed line) may deviate from the ground-truth focus (black line), especially when the peak is flat or locally irregular. In contrast, the weighted centroid (red line), computed over $\Omega$ (shaded region), consistently aligns more closely with the ground truth by aggregating stable evidence across neighboring blocks.

This aggregation-based localization avoids brittle single-point decisions and provides stable focus estimates across long focus sweeps and dynamic scenes.

*Table 9.* Focus localization quality measured by GT ratio (higher is better) and GT rank (lower is better), shown as Ratio / Rank. Best is bold; second best is underlined (ties handled conservatively).

| Scene | GRAD | LAP | SD | COUNT | MFDCT | HF | CEN |
|---|---|---|---|---|---|---|---|
| USAF_static_constant | 0.177 / 1132 | 0.030 / 1439 | 0.986 / 8 | 0.144 / 1462 | 0.107 / 1320 | 0.949 / 13 | **0.988 / 4** |
| USAF_static_decrease | 0.940 / **18** | 0.516 / 840 | **0.978** / 25 | 0.243 / 626 | 0.774 / 195 | 0.974 / 29 | 0.978 / 42 |
| USAF_static_increase | 0.874 / 23 | 0.636 / 413 | 0.993 / 9 | 0.203 / 1083 | 0.975 / 5 | **0.995 / 3** | 0.996 / 4 |
| USAF_static_minor | 0.619 / 312 | 0.961 / 27 | **0.995 / 2** | 0.011 / 1518 | 0.312 / 1080 | 0.968 / 24 | 0.983 / 19 |
| USAF_dynamic_constant | **1.000 / 1** | 0.981 / 5 | 0.987 / 17 | 0.161 / 1206 | **1.000 / 1** | 0.996 / 2 | **1.000 / 1** |
| USAF_dynamic_decrease | 0.763 / 9 | 0.408 / 673 | 0.983 / 14 | 0.393 / 769 | 0.378 / 5 | **0.994** / 6 | 0.991 / 11 |
| USAF_dynamic_increase | 0.981 / **8** | 0.503 / 526 | 0.985 / 17 | 0.292 / 972 | 0.754 / 100 | 0.981 / 18 | **0.987** / 23 |
| USAF_dynamic_minor | 0.180 / 1258 | 0.609 / 669 | **0.988 / 6** | 0.025 / 1508 | 0.081 / 1443 | 0.959 / 42 | 0.972 / 33 |
| USAF_dynamic_significant | 0.059 / 1469 | 0.846 / 254 | 0.977 / 44 | 0.025 / 1428 | 0.049 / 1497 | 0.985 / **5** | **0.991** / 7 |
| Bottle_static_constant | 0.991 / 3 | 0.964 / 6 | 0.983 / 3 | 0.470 / 138 | 0.984 / 2 | 0.993 / 9 | **0.996 / 3** |
| Bottle_dynamic_constant | 0.974 / 6 | 0.981 / 4 | **1.000 / 1** | 0.441 / 202 | 0.959 / 4 | 0.956 / 11 | 0.976 / 9 |
| Lily_static_constant | 0.883 / 26 | 0.617 / 319 | 0.945 / 45 | 0.403 / 507 | 0.797 / 53 | **0.945 / 24** | 0.924 / 27 |
| Lily_dynamic_constant | 0.786 / 107 | 0.713 / 276 | **0.927** / 55 | 0.447 / 559 | 0.762 / 60 | 0.847 / **45** | 0.921 / 48 |
| Fan_static_constant | 0.926 / 4 | 0.656 / 58 | **1.000 / 2** | 0.377 / 79 | 0.986 / 2 | 0.798 / 23 | 0.996 / 3 |
| Fan_dynamic_constant | 0.610 / 74 | 0.637 / 76 | 0.957 / 8 | 0.208 / 122 | 0.718 / 8 | 0.563 / 68 | **0.997 / 2** |
| MEAN | 0.717 / 296.67 | 0.670 / 372.33 | **0.979** / 17.07 | 0.256 / 811.93 | 0.642 / 385 | 0.927 / 21.47 | 0.980 / **15.73** |

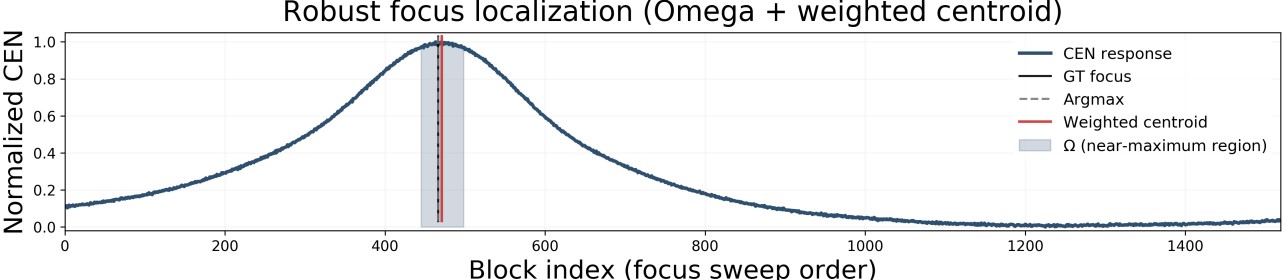

*Figure 7.* Illustration of robust focus localization using near-maximum aggregation. The normalized CEN response is shown over the focus sweep. The shaded region indicates the near-maximum set $\Omega$, defined by a tolerance around the global maximum. Vertical lines mark the ground-truth focus (black), the argmax prediction (gray dashed), and the proposed weighted centroid estimate (red). By aggregating evidence across $\Omega$, the centroid-based localization avoids brittle single-point decisions and yields more accurate and stable focus estimates under flat or irregular peaks.

*Table 10.* Parameter sensitivity analysis on real spike-camera sequences. Mean and median absolute error ranges are reported in block units.

| Parameter | Values tested | Mean Err. | Median Err. |
|---|---|---|---|
| $\alpha_{\mathrm{prom}}$ | 0.55, 0.80, **1.10**, 1.40, 1.65 | 2.73–3.20 | 2.00–2.00 |
| $\alpha_{\mathrm{curv}}$ | 0.10, 0.20, **0.35**, 0.50, 0.70 | 2.80–2.80 | 2.00–2.00 |
| $\alpha_{\mathrm{width}}$ | 0.15, 0.30, **0.45**, 0.60, 0.75 | 2.73–3.13 | 2.00–2.00 |
| $\alpha_{\mathrm{plat}}$ | 0.10, 0.20, **0.35**, 0.50, 0.70 | 2.73–2.93 | 2.00–2.00 |
| $\alpha_{\mathrm{edge}}$ | 0.20, 0.40, **0.60**, 0.80, 1.00 | 2.80–33.07 | 2.00–2.00 |
| $\tau$ | 0.005, 0.008, **0.012**, 0.016, 0.020 | 2.80–4.13 | 2.00–3.00 |
| $\gamma$ | 1.0, 1.5, **2.0**, 3.0, 4.0 | 2.80–2.80 | 2.00–2.00 |

## A.3. Parameter sensitivity.

We further analyze the sensitivity of CEN to its fixed hyperparameters, including the five weights in the structure-consistency score, the near-maximum tolerance $\tau$, and the exponent $\gamma$ in Eq. 14. For each parameter, we vary its value while keeping all

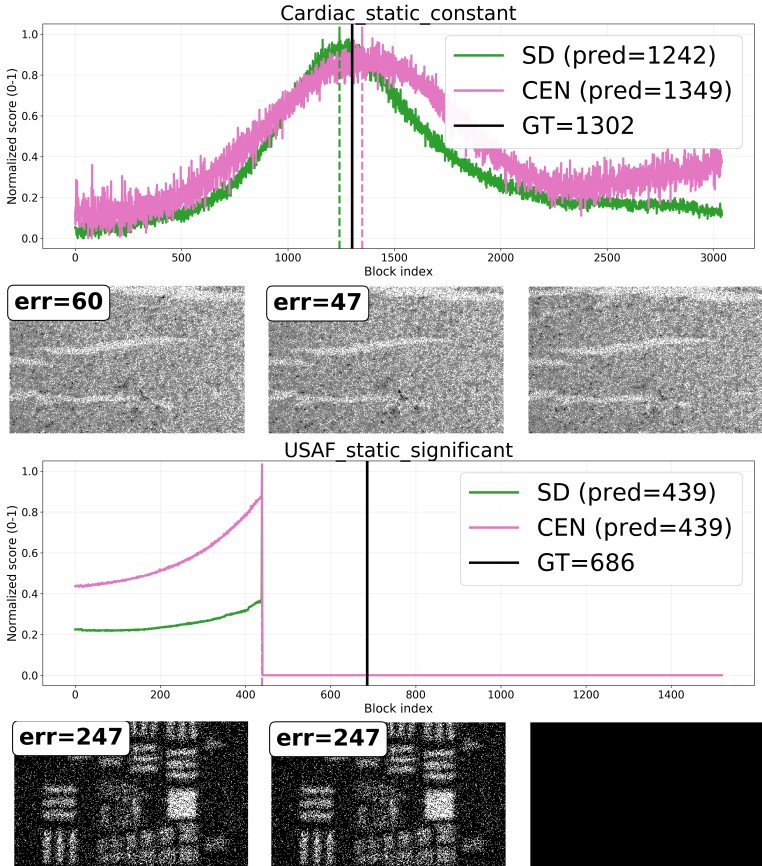

*Figure 8.* Representative failure cases. Top: an extremely noisy scene where noise dominates the response tail after the GT focus position. Bottom: the USAF static significant scene, where illumination drops to near zero and no spikes are generated at the GT focus. For each case, response curves of SD and CEN are shown together with the GT, and the accumulated spike blocks at the predicted focus are visualized with absolute block error indicated.

other parameters fixed, and report the range of mean and median absolute errors over all real spike-camera sequences.

As shown in Table 10, CEN is generally insensitive to these hyperparameters. Across a broad range of values, the median error remains stable at 2 blocks for almost all settings, and the mean error changes only mildly. The only noticeable exception is $\alpha_{\text{edge}}$ when set too small: weakening the edge penalty may allow boundary-dominated extrema to be selected in rare cases, leading to a large mean error due to an outlier, while the median error remains unchanged. In practice, all parameters are fixed once and applied uniformly across sequences without per-scene adjustment, indicating that the performance gain does not come from dataset-specific parameter tuning.

### A.4. Failure cases.

We analyze representative failure cases in Figure 8. The first case is an extremely noisy scene, where high-frequency noise dominates the spectrum after the ground-truth focus position. Although defocus should reduce high-frequency content, the observed rising tail indicates noise dominance, causing CEN to select an inappropriate radius $r$ and bias the estimated focus toward the noisy region. Nevertheless, this scene remains challenging for all methods, and even the best baseline (SD) yields larger errors than CEN.

The second case is the *USAF static significant* scene, where illumination drops to near zero at the ground-truth focus. Under such conditions, no spikes are generated, and no focus-related information is available. All methods fail in this scenario, reflecting an inherent limitation of event- and spike-based sensors under zero-illumination conditions rather than a deficiency of CEN.

