# OpenReview forum: "Spike Camera Autofocus via Frequency-Domain Spectral-Centroid Migration"
_ICML.cc/2026/Conference — ICML 2026 regular_

### Official Review · Reviewer_AzEy · 2026-03-10

**Soundness:** 2
**Presentation:** 3
**Significance:** 2
**Originality:** 2
**Overall Recommendation:** 4
**Confidence:** 3

**Summary:**

The paper addresses the challenge of autofocusing spike cameras, which produce sparse, binary event streams that lack the conventional intensity gradients used in standard autofocus systems. The authors observe that sweeping the focal plane induces a reliable migration of spectral energy in the spatial-frequency domain. Leveraging this physical property, they propose CEN (Centroid-based Energy Navigation), a method that tracks the spectral centroid of accumulated spike blocks. To ensure robustness against noise and irregular signal responses, the method utilizes a structure-consistent identification mechanism to dynamically select the optimal frequency bound (radius), followed by a weighted aggregation step for stable focus localization. The proposed method operates entirely in the frequency domain, circumventing the need for intermediate image reconstruction, and demonstrates strong empirical performance across various motion and illumination conditions on the SAD dataset.

**Compliance With Llm Reviewing Policy:**

Affirmed.

**Final Justification:**

The authors' rebuttal and subsequent discussion successfully addressed my primary concerns. The addition of the continuous optical defocus derivation provides a much-needed theoretical foundation, and the new reconstruction baselines (TFP/TFI + RDF/DRDF) clearly validate the advantage of operating directly in the frequency domain.

Crucially, the authors have explicitly committed to documenting the method's real-world constraints in a dedicated Limitations section for the camera-ready version, specifically, the boundary-failure vulnerability tied to the $\alpha_{edge}$ hyperparameter and the significant hardware overhead of running $O(N \log N)$ FFTs on edge devices compared to $O(N)$ temporal metrics.

Given the strengthened theoretical grounding and the commitment to scientific transparency regarding its operational boundaries, I have raised my score to a Weak Accept.

**Key Questions For Authors:**

- In Table 1, there is a severe and unexplained performance variance for the HF (high-frequency energy) baseline. HF achieves state-of-the-art or near state-of-the-art performance on the USAF (3.33) and Lily (10.50) sequences, yet it collapses entirely on the Fan sequence (88.50). What specific scene characteristics (e.g., texture frequency distributions, motion profiles) drive this catastrophic failure mode in HF, and mathematically, how does CEN's structural selection guarantee stability in these exact conditions?

- The pipeline relies on accumulating spikes over a fixed window of $d_t$ frames. In the presence of rapid intra-block scene motion, motion blur will attenuate high-frequency spectral components in a manner mathematically identical to optical defocus. How does CEN fundamentally disentangle spectral shifts caused by true lens defocus from those caused by severe dynamic motion?

- The structure-consistency score (Eq. 7) utilizes five fixed empirical weights (e.g., 1.10, 0.35, 0.60) applied uniformly. This strongly implies the method is overfit to the specific focus sweep characteristics and noise profiles of the SAD dataset. Can the authors provide a rigorous sensitivity analysis or results on a completely unseen, out-of-domain dataset to prove these exact heuristic weights generalize?

- Neuromorphic vision sensors are typically deployed in constrained edge environments where microsecond latency and microjoule power consumption are critical. What is the actual wall-clock latency and energy cost of computing continuous FFTs on edge hardware compared to simple temporal counters like Spike Dispersion?

**Limitations:**

The authors have not adequately discussed the limitations of their work. While they highlight failure cases related to zero-illumination and extreme noise, they fail to address the fundamental limitations of their core methodology. The authors must explicitly discuss the generalization limits of their highly parameterized heuristic scoring system (Eq. 7). Furthermore, they must include a candid discussion on the severe computational latency and power trade-offs of relying on continuous frequency-domain transformations for real-time neuromorphic systems, as this directly impacts real-world deployability.

**Strengths And Weaknesses:**

Strengths:

The paper is clear, and easy to follow. Formulating spike autofocus as a frequency-domain statistical problem is mathematically sound and physically intuitive. The pipeline effectively bypasses the artifacts and latency associated with intermediate image reconstruction. While spike camera autofocus is a specialized domain, this work provides a meaningful shift away from purely temporal or spatial heuristics (like Spike Dispersion) toward a more principled, physically grounded metric. Applying spectral analysis to focus evaluation is a classical concept in frame-based imaging; however, adapting it strictly for sparse spike streams using an adaptive bounded centroid and structure-consistent selection represents a novel and well-justified combination of techniques.

Weaknesses:

1. The methodology relies heavily on accumulating spikes over $d_t$ frames to form evaluation blocks (Eq. 1), which assumes strict scene stability. In dynamic environments, rapid intra-block motion will cause motion blur that attenuates high frequencies—perfectly mimicking defocus blur. The proposed spectral centroid metric cannot fundamentally disentangle these two phenomena. Furthermore, the structure-consistency score (Eq. 7) is highly problematic. It relies on a linear combination of five hardcoded hyperparameters ($\alpha$) tuned uniformly across the dataset. This heavy parameterization severely undermines the method's robustness and raises significant concerns regarding overfitting.

2. The paper lacks formal theoretical justification for the specific functional forms chosen for the structural consistency penalties. The method reads as a collection of engineered heuristics rather than a principled mathematical derivation.

3. The practical utility of this approach is fundamentally bottlenecked by its computational overhead. Executing continuous, block-wise 2D Fourier transforms ($O(N \log N)$) imposes substantial processing and energy demands, actively negating the ultra-low-power, low-latency advantages that justify the use of neuromorphic edge sensors in the first place.

4. Utilizing spectral analysis and high-frequency energy migration for focus evaluation is a textbook concept in conventional frame-based computational imaging. Because the core foundation is entirely standard, the paper's novelty hinges on the adaptation mechanics (the heuristic selection in Eq. 7), rendering the scientific contribution highly incremental.

---

> ### Author Rebuttal · Authors · 2026-03-28
>
> We thank the reviewer for the constructive feedback and insightful comments.
>
> ---
> ## **Concern 1: Motion blur vs. defocus blur**
>
> We agree that both motion and defocus attenuate high-frequency components, and may appear similar at a single observation.
>
> However, CEN does not rely on a single spectral snapshot, but on the evolution of the spectral centroid across the focus sweep. Defocus is coupled to lens position and induces a consistent outward-and-return migration, whereas motion blur is not sweep-aligned and does not produce such structured evolution. Thus, while locally similar, they differ in their global trajectory, which is the key signal exploited by CEN.
>
> Results on dynamic scenes (Table 3) show that, even when motion and defocus coexist, the proposed method can still identify this migration and localize focus.
>
> CEN does not attempt to distinguish all motion-induced blur, but exploits consistency when such structure is present. Under extremely strong motion, where this pattern is disrupted, performance may degrade. We will clarify this limitation.
>
> ---
> ## **Concern 2: Severe variance of the HF baseline**
> We thank the reviewer for this question.
>
> In the Fan sequence, we observe periodic burst-like spike activity. This introduces abnormal high-frequency energy even under strong defocus, leading to spurious maxima. This leads to high-frequency responses that are not aligned with focus-induced spectral migration.
>
> CEN mitigates this via structure-consistent selection, favoring curves with a clear interior extremum and suppressing irregular responses. We will clarify this in the revision.
>
> ## **Concern 3: Parameterization, overfitting, lack of theory**
> We thank the reviewer for this concern.
>
> Eq. (7) is not a heuristic objective but a selection mechanism that recovers the response best aligned with spectral-centroid migration. Importantly, the core principle of the method is defined by spectral-centroid migration, while Eq. (7) only serves to select the most consistent realization of this principle in practice.
>
> The parameters do not define the focus criterion, but select among candidate responses generated by the spectral principle. This principle follows standard imaging behavior: defocus attenuates high frequencies, shifting spectral energy and inducing a rise-and-fall centroid response.
>
> The terms act as structural proxies:
> - prominence enforces a clear extremum
> - curvature and width favor localized peaks;
> - plateau and edge suppress flat responses or boundary artifacts
>
> This establishes a direct link between optical defocus and the observed spectral-centroid behavior.
>
> From this perspective, the formulation encodes structural preferences rather than dataset-specific numerical patterns. The parameters regulate the relative importance of these structural properties, rather than fitting data.
>
> This is supported by the sensitivity analysis (please see response 1 to Reviewer rwQi), where performance remains stable across a broad range of values, indicating that these parameters act as soft constraints rather than precise tuning.
>
> We will clarify this interpretation more explicitly in the revision.
>
> ---
> ## **Concern 4: Computational overhead**
> We agree that CEN is slightly more expensive than comparisons due to FFT, while it remains the same order of magnitude as other frequency-based methods.
>
> Although this work does not specifically optimize for hardware latency or energy efficiency. Prior studies on FPGA-based FFT implementations have demonstrated microsecond-level latency [1] and real-time processing capability [2], indicating that the core spectral computation in CEN is compatible with efficient deployment on appropriate hardware.
>
> We thank the reviewer for pointing this out, which highlights an important direction for future improvement. We will clarify this limitation in the revision.
>
> **References:**
>
> [1] Efficient FPGA Implementation of FFT/IFFT Processor.
>
> [2] FPGA Implementations of Fast Fourier Transforms for Real-Time Signal and Image Processing.
>
> ---
> ## **Concern 5: Novelty**
> While spectral analysis is classical in frame-based autofocus, our contribution is to identify a spike-observable focus cue and reformulate spike-camera autofocus as a frequency-domain problem, as noted by reviewers (KTXp, vsPL). Specifically, we make spectral energy migration observable in spike data via a bounded spectral centroid, enabling frequency-domain autofocus without reconstruction. This transforms autofocus from a spatial or temporal heuristic into a frequency-domain representation problem under a new sensing modality.
>
> ---
> ## **Concern 6: Limitations**
> We thank the reviewer for this suggestion. We will explicitly state that:
>
> (i) the current implementation has not yet been evaluated on edge hardware;
>
> (ii) although Eq. (7) is robust in our sensitivity analysis and fixed across all scenes, broader cross-dataset and cross-sensor validation remains important to further assess generalization.

---

> > ### Author Rebuttal · Reviewer_AzEy · 2026-04-01
> >
> > Thank you for the detailed rebuttal and responses to the other reviewers. While my initial review raised major concerns about heuristic overfitting and computational overhead, the new evidence has prompted a reassessment.
> >
> > The continuous optical defocus derivation (provided to Reviewer vsPL) strengthens the paper's theoretical foundation. Additionally, the comparisons against TFP/TFI reconstructed baselines with RDF/DRDF (provided to Reviewer KTXp) clearly show the artifacts caused by intermediate reconstruction, validating your frequency-domain approach.
> >
> > As a strong paper requires scientific transparency about its limitations, two critical concerns must be explicitly documented in the manuscript:
> >
> > 1- Hyperparameter Sensitivity ($\alpha_{edge}$):
> > You note to Reviewer rwQi that the method is "not overly sensitive." However, the data shows the edge penalty ($\alpha_{edge}$) is a critical safeguard. Setting $\alpha_{edge}$ to 1.00 causes the mean error to spike to 33.07 due to a boundary failure on the USAF_dynamic_increase sequence. This confirms the method is brittle to boundary-dominated extrema unless this parameter is precisely tuned. This vulnerability must be explicitly stated, not masked by pointing to median stability.
> >
> > 2- Hardware Constraints vs. FFT Overhead:
> > Citing a general-purpose FPGA FFT paper does not address the system-level bottleneck. Neuromorphic edge sensors operate under strict microsecond/microjoule constraints. A block-wise 2D FFT ($O(N \log N)$) requires vastly more memory bandwidth and compute than simple $O(N)$ temporal metrics like Spike Dispersion.
> >
> > Follow-up Question:
> >
> > Will you commit to including a dedicated "Limitations" section in the camera-ready manuscript that explicitly addresses:
> >
> > (a) The system's vulnerability to sweep-boundary failures if the $\alpha_{edge}$ heuristic is not tuned to the sweep length?
> >
> > (b) The relative hardware complexity, latency, and memory overhead of running continuous $O(N \log N)$ FFTs on edge processors versus $O(N)$ temporal metrics?
> >
> > If you agree to integrate these limitations to ensure the method's constraints are accurately reflected in the literature, I will raise my score to a Weak Accept.

---

> > > ### Author Response · Authors · 2026-04-02
> > >
> > > We sincerely thank the reviewer for the careful follow-up and for reconsidering the paper with the additional theoretical and experimental evidence.
> > >
> > > We agree that these limitations should be stated explicitly in the manuscript, and **we will explicitly address these points in the Limitations section of the camera-ready version.**
> > >
> > > In particular, we will state the following two points:
> > >
> > > **(1) Sensitivity of $\alpha_{\mathrm{edge}}$ and vulnerability to boundary failures.**
> > > We agree that the sensitivity analysis shows the edge term $\alpha_{\mathrm{edge}}$ is an important safeguard against boundary-dominated extrema. Although the method is generally stable across a broad range of settings, when $\alpha_{\mathrm{edge}}$ is not properly set, the method may select incorrect extrema and lead to rare but failure cases, as observed in the USAF\_dynamic\_increase sequence.  In the revised manuscript, we will explicitly state that $\alpha_{\mathrm{edge}}$ must be set consistently with the sweep configuration, as improper settings may lead to boundary-selection failures.
> > >
> > > **(2) System-level hardware overhead of FFT-based processing.**
> > > Compared with lightweight $O(N)$ temporal metrics such as Spike Dispersion, CEN relies on repeated block-wise 2D FFTs and therefore involves higher computational cost and additional memory access due to $O(N \log N)$ processing.
> > > As a result, although CEN enables more accurate focus estimation and improved robustness, it has a less favorable hardware profile than simple temporal counters in ultra-low-latency and energy-constrained edge settings. We will make this limitation explicit in the manuscript and clarify that this trade-off should be considered when evaluating the method for real-time neuromorphic deployment.
> > >
> > > We thank the reviewer again for highlighting these important points. We agree that making these limitations explicit will improve the paper’s scientific clarity and ensure that its practical constraints are accurately reflected.

---

### Official Review · Reviewer_vsPL · 2026-03-12

**Soundness:** 3
**Presentation:** 3
**Significance:** 3
**Originality:** 3
**Overall Recommendation:** 4
**Confidence:** 4

**Summary:**

This paper studies the autofocus problem of spike cameras. Due to the sparse and binary nature of spike data, traditional focus evaluation functions based on image sharpness perform poorly. The authors propose the CEN method, whose core observation is that during the focusing scan, spectral energy migrates towards higher frequencies in the spatial frequency domain and falls back when the image is out of focus. Based on this phenomenon, CEN constructs a focus curve by calculating a finite spectral centroid, automatically selects the frequency radius using structural consistency scoring, and finally locates the focus using a weighted near-maximum. Experiments show that on the SAD dataset, this method outperforms traditional operators such as GRAD and LAP, as well as recent SD methods.

**Compliance With Llm Reviewing Policy:**

Affirmed.

**Key Questions For Authors:**

This paper studies a novel problem, and I am inclined to give it a positive score. I hope the authors can address the questions above during the rebuttal; if they do, I will maintain my positive evaluation.

**Strengths And Weaknesses:**

Advantages

Strong Problem Targeting: This method specifically addresses the problem of traditional focus functions failing due to the sparsity of spike camera data.

Novel Approach: Utilizing the phenomenon of spectral energy migration during focusing to construct an evaluation function is rarely systematically studied in spike camera scenarios.

Simple and Efficient Method: The method requires no image reconstruction, only frequency domain statistics, resulting in low computational cost and potential for practical application.

The experiments are relatively complete: Multiple index comparisons, robustness analyses, and several ablation experiments are provided under a given benchmark.

Weaknesses

Narrow experimental validation scope: All results are derived from a single SAD dataset, lacking validation under different sensor, lens systems, or scene conditions; therefore, generalization ability remains unclear.

Excessive heuristic design in the methodology: The structural consistency score includes multiple manually set weights and hyperparameters, but the paper does not provide sufficient parameter sensitivity analysis or explain the basis for parameter selection.

Insufficient theoretical analysis: The key assumption that the spectral centroid exhibits a single-peak trend with focus variation is primarily based on empirical observation, lacking theoretical explanation based on optical imaging models or sensor models.

Insufficient benchmark comparison: The paper emphasizes the absence of reconstruction, but does not compare methods that "reconstruct pulse data first, then use a mature focus evaluation function," a necessary benchmark to support the paper's conclusions.

Performance improvements are not consistent: In some scenarios, traditional methods perform close to or better than CEN; the paper discusses these situations in limited detail.

The paper's expression still needs improvement: for example, the number of terms in formula (7) is inconsistent with the textual description, and some experimental tables (such as Table 5) lack units or hardware information, which affects the reproducibility of the method.

---

> ### Author Rebuttal · Authors · 2026-03-28
>
> We thank the reviewer for the careful reading, positive evaluation, and constructive feedback.
>
> ---
>
> ## **Concern 1: Narrow experimental validation scope**
> We thank the reviewer for the constructive and valuable feedback.
>
> Our goal is not to claim full cross-sensor generalization, but to establish a sensor-level principle that is expected to transfer across settings.
>
> Importantly, our formulation is based on a sensor-observable frequency-domain property induced by defocus, rather than dataset-specific statistics or learned parameters. This distinguishes it from methods that rely on data-dependent tuning.
>
> Within the SAD benchmark, the consistent performance across diverse motion patterns and scene conditions (Tables 3–5) provides initial evidence that the proposed cue is not tied to a specific configuration.
>
> We agree that cross-sensor and cross-optics validation is an important next step. This comment is valuable and will guide our future work to further evaluate the method on additional spike-camera platforms and imaging setups.
>
> ---
>
> ## **Concern 2: Heuristic design / hyperparameters**
>
> We respectfully refer the reviewer to our responses to Reviewer rwQi (Weakness 1), where we provide detailed parameter sensitivity analysis and clarify the basis for parameter selection.
>
> ---
>
> ## **Concern 3: Insufficient theoretical analysis**
>
> We thank the reviewer for the constructive suggestion. Following this comment, we have supplemented the paper with a clearer optical-model-based explanation for why the spectral-centroid response tends to peak near best focus.
>
> Our analysis starts from a standard defocus imaging model, where the observed signal is formed by convolving a latent sharp image with a defocus point spread function (PSF). Let $I(x)$ denote the latent sharp signal and $h_{\sigma}(x)$ the PSF associated with blur scale $\sigma$. The blurred observation is
> $$I_{\sigma}(x) = (I * h_{\sigma})(x).$$
> In the frequency domain,
> $$P_{\sigma}(\omega)=|\widehat{I}(\omega)|^2\,|\widehat{h_{\sigma}}(\omega)|^2.$$
> For typical defocus, $|\widehat{h_{\sigma}}(\omega)|^2$ behaves as a low-pass filter. Using a Gaussian approximation,
> $$P_{\sigma}(\omega)=|\widehat{I}(\omega)|^2\exp\left(-\sigma^2\|\omega\|^2\right),$$
> which shows that higher frequencies are increasingly suppressed as $\sigma$ grows.
>
> This directly motivates our centroid formulation in Eq. (5). In continuous form,
> $$C_r(\sigma)=\frac{\int_{\|\omega\|\le r}\|\omega\|\,P_{\sigma}(\omega)\,d\omega}{\int_{\|\omega\|\le r}P_{\sigma}(\omega)\,d\omega+\varepsilon}.$$
>
> During a focus sweep, $\sigma$ decreases when approaching focus and increases afterward, causing spectral energy to first shift outward and then inward. As a weighted average of frequency, the centroid therefore exhibits an increase-then-decrease trend, leading to a peak near the best-focused block.
>
> In practice, spike measurements are affected by noise, quantization, motion, and illumination variation, which may distort this ideal trend. This is precise with the further introduced structure-consistency criterion in Eq. (7), which is designed to identify the response that best matches the underlying focus-induced spectral migration under such conditions.
>
> We will incorporate this theoretical explanation into the revised manuscript.
>
> ---
>
> ## **Concern 4: Missing reconstruction-based baselines**
>
> We have conducted additional experiments following this protocol (reconstruct first, then apply image-based focus measures), and we respectfully refer the reviewer to our response to Reviewer KTXp (Weakness 2) for details.
>
> ---
>
> ## **Concern 5: Performance improvements are not always consistent**
>
> We thank the reviewer for this observation. We acknowledge that the current discussion of these cases can be further improved.
>
> In some scenarios, traditional methods can perform comparably when their underlying assumptions (e.g., regular textures or stable conditions) are well satisfied. For example, in the USAF scene (Table 1), HF achieves slightly lower error than CEN, and SD performs well in static settings (Table 2). However, their performance becomes less reliable outside these conditions. In dynamic scenes such as Fan, HF is affected by burst-like spike activity, while SD degrades under motion, whereas CEN remains accurate by capturing a global frequency-domain migration pattern.
>
> We will expand the discussion in the revised manuscript to more clearly analyze these cases and explain the observed performance differences.
>
> ---
>
> ## **Concern 6: Writing issues and reproducibility**
>
> We thank the reviewer for pointing out these issues.
>
> Eq. (7) indeed consists of **five** terms, and we will correct the textual description to ensure consistency.
>
> We will also explicitly report units (e.g., ms per block) and include hardware details for all experimental tables to improve clarity and reproducibility. In addition, we will release the code to further facilitate reproducibility.

---

> > ### Author Rebuttal · Reviewer_vsPL · 2026-04-05
> >
> > Thanks for your detailed rebuttal; I will maintain my positive score.

---

### Official Review · Reviewer_rwQi · 2026-03-12

**Soundness:** 4
**Presentation:** 3
**Significance:** 3
**Originality:** 4
**Overall Recommendation:** 5
**Confidence:** 4

**Summary:**

This paper proposes a frequency-domain autofocus method CEN for spike cameras, based on the observation that spatial-frequency energy migrates toward high frequency as the scene comes into focus. The method defines a bounded spectral centroid on spike-accumulated blocks, selects a structure consistent radius via a curve-shape score, and then localizes the focus using a robust centroid. Experiments on the SAD dataset suggest that CEN improves accuracy and robustness compared with existing methods, and the paper also includes ablations and failure-case analysis.

**Compliance With Llm Reviewing Policy:**

Affirmed.

**Key Questions For Authors:**

see weaknesses

**Limitations:**

yes

**Strengths And Weaknesses:**

Strengths
1. The pipeline looks reasonable. The definitions of the spectral centroid, the radius selection criterion, and the robust focus localization are precise enough to implement. Experiments are run on a unified benchmark with shared settings for all baselines, and the metrics are appropriate. The ablations are helpful to justify each design choice.
2. The paper is generally clear and easy to follow. The motivation is concrete, the main algorithm is summarized nicely, and the experimental section is well structured.
3. For the spike-camera autofocus task, the contribution is quite meaningful. It directly improves a focus control and is relatively simple and efficient to deploy. The paper also offers a new perspective that the proposed method can work
directly in the spectral domain without any image reconstruction.

Weaknesses
1. The method uses several fixed hyperparameters (weights in the structure consistency score, tolerance for the near-maximum region, exponent γ). Could you add a paragraph or table showing that the performance is not overly sensitive to these choices, or explaining how they were selected in practice?
2. For scenes with multiple depth layers or more complex motion, like both camera and objects moving, how do you expect the spectral-centroid curve to behave? A short discussion on such a challenging case would help clarify the applicability limits of the method.
3. The method relies on the observation that spectral energy shifts toward higher frequencies when approaching focus. How about the scenarios where texture or noise also introduces strong high-frequency components? How the method distinguishes such cases from true focus changes?

---

> ### Author Rebuttal · Authors · 2026-03-27
>
> We sincerely thank the reviewer for the positive and thoughtful assessment.
>
> ---
>
> ## **Weakness 1: Parameter sensitivity**
>
> We thank the reviewer for this helpful suggestion.
> Following this comment, we conducted a *sensitivity analysis* covering all fixed hyperparameters in our method, including the five weights in the structure-consistency score, the tolerance $\tau$ and the exponent $\gamma$.
>
> **Parameter sensitivity results**
>
> | Parameter | Values tested | Mean abs. error range | Median range |
> |---|---|---:|---:|
> | $\alpha_{\mathrm{prom}}$ | 0.55, 0.80, **1.10**, 1.40, 1.65 | 2.73--3.20 | 2.00--2.00 |
> | $\alpha_{\mathrm{curv}}$ | 0.10, 0.20, **0.35**, 0.50, 0.70 | 2.80--2.80 | 2.00--2.00 |
> | $\alpha_{\mathrm{width}}$ | 0.15, 0.30, **0.45**, 0.60, 0.75 | 2.73--3.13 | 2.00--2.00 |
> | $\alpha_{\mathrm{plat}}$ | 0.10, 0.20, **0.35**, 0.50, 0.70 | 2.73--2.93 | 2.00--2.00 |
> | $\alpha_{\mathrm{edge}}$ | 0.20, 0.40, **0.60**, 0.80, 1.00 | 2.80--33.07 | 2.00--2.00 |
> | $\tau$ | 0.005, 0.008, **0.012**, 0.016, 0.020 | 2.80--4.13 | 2.00--3.00 |
> | $\gamma$ | 1.0, 1.5, **2.0**, 3.0, 4.0 | 2.80--2.80 | 2.00--2.00 |
>
> Overall, the method is not overly sensitive to these hyperparameters. Across a broad range of values, the performance remains stable: the median absolute focusing error stays at 2 blocks for almost all settings, and the mean error varies only slightly. In particular, $\alpha_{\mathrm{curv}}$ and $\gamma$ have negligible impact, while the other parameters introduce only mild variations.
>
> We observe that $\alpha_{\mathrm{edge}}$ has a more visible effect when set to a very small value. For example, when $\alpha_{\mathrm{edge}}=0.2$, the mean error increases due to a single extreme outlier (USAF\_dynamic\_increase, error = 455), while the median remains unchanged and all other scenes stay stable. This is consistent with the role of the edge term, which suppresses spurious extrema near sweep boundaries; weakening it allows occasional boundary-dominated failures.
>
> In practice, all parameters are fixed once and applied uniformly across sequences, without per-scene adjustment. This indicates that the parameters are not used to fit dataset-specific patterns.
>
> We will include a sensitivity table in the revision to further support this robustness.
>
> ---
>
> ## **Weakness 2: Multi-depth and complex motion scenarios**
>
> We thank the reviewer for this important question.
>
> In multi-depth scenes, the spectral-centroid curve reflects a superposition of responses from different depth layers.
> When one depth contributes more spike responses, the overall curve is still dominated by that layer and preserves a clear and structured evolution across the focus sweep.
>
> When multiple depths contribute comparably, the curve can become less sharp, with a flatter or less localized transition region. In such cases, the focus indication becomes less distinct, which may reduce accuracy.
>
> For complex motion (e.g., simultaneous camera and object motion), motion introduces additional frequency components into the spectrum. Unlike defocus, these components do not follow a consistent inward or outward migration pattern across the focus sweep. As a result, they tend to produce less stable or less structured variations in the spectral-centroid curve.
>
> This difference is what our structure-consistency modeling leverages: it favors regions with stable and consistent spectral evolution, which helps suppress motion-induced interference in practice. Nevertheless, in extreme cases with strong and irregular motion, the curve can still be distorted, which remains a challenging scenario.
>
> We will clarify these applicability limits in the revised manuscript.
>
> ---
>
> ## **Weakness 3: High-frequency texture/noise versus true focus changes**
>
> We agree that strong texture or noise may also produce high-frequency components.
>
> To clarify, CEN does not rely on the magnitude of high-frequency energy at a single focus step. Instead, it leverages the *migration pattern* of the bounded spectral centroid across the focus sweep. True focus change tends to induce a more structured and persistent outward-and-return evolution, whereas noise or incidental texture typically leads to less consistent or less localized response variations.
>
> This difference motivates our use of structure-consistent response identification, rather than directly relying on a fixed-band high-frequency statistic. As a result, the method is generally less sensitive to incidental high-frequency fluctuations. Our experimental results support this observation: the HF baseline can perform well in some scenes but fails badly in others, while CEN remains much more stable overall (Table 1).
>
> The radius-selection ablation and the within-$r$ visualization further show that proper truncation helps suppress distracting high-frequency residuals outside the focus-discriminative band.
>
> We will clarify this explanation more explicitly in the revision.

---

> > ### Author Rebuttal · Reviewer_rwQi · 2026-04-04
> >
> > It addressed most of my concerns. And I have no further questions about this work. Due to the discussion with the reviewers, I think I need to point out that spike focus is one direction of great interest in neuromorphic learning, fitting the topic requirements of ICML. I will maintain my score.

---

### Official Review · Reviewer_KTXp · 2026-03-13

**Soundness:** 3
**Presentation:** 2
**Significance:** 1
**Originality:** 2
**Overall Recommendation:** 2
**Confidence:** 3

**Summary:**

This paper introduces a frequency domain autofocus method for spike cameras. Its key contribution is the observation that, during a focus sweep, the spectral energy in spike data shifts outward and then inward, and the best focus can be found from this pattern. Based on that, the authors design CEN, a focus measure that tracks spectral centroid movement and uses it to estimate the correct focus position directly from spike signals, without reconstructing images. They then show that this works better than several existing autofocus baselines on the spike autofocus benchmark.

**Compliance With Llm Reviewing Policy:**

Affirmed.

**Final Justification:**

I thank the authors for their responses. However, I believe the latest responses make the case for the paper weaker rather than stronger. I detail my reasons below.

1. The paper does not feature any learning component, nor does it clearly include a part that contributes directly to machine learning. The authors themselves state, *“We agree that our method is not a learning-based model in the narrow sense.”*

2. The authors argue that their algorithm extracts meaningful structure from data. In that logic, however, many algorithms, including those far removed from machine learning, could also be framed as machine learning contributions. I do not find this argument convincing. The comparison to the ICML 2005 paper is also weak and somewhat misleading. The cited paper, *Proto-Value Functions: Developmental Reinforcement Learning* (ICML 2005), is clearly tied to reinforcement learning and representation learning. The authors did not really address this point.

3. The added MLP experiment does not appear to establish a meaningful machine learning contribution. Since the student model uses hand-designed radial spectral features rather than raw spatial inputs, the experiment mainly shows that CEN can provide a somewhat better pseudo-supervisory signal than a naive center prior in a limited controlled setting. This is a weak claim rather than evidence of a CEN-based end-to-end learning model. It does not change the fact that the paper’s main contribution remains a handcrafted autofocus measure rather than a learning-based method.

4. While I appreciate the additional synthetic data results, I believe this also raises further questions. The authors explicitly state that they followed the same evaluation protocol as the main paper, yet the main paper reports relative absolute error, whereas the new synthetic results are suddenly reported in meters. This makes me question why the same metric from the SAD dataset was not used.

5. Another point I noticed after my review is that the paper evaluates only four scenes from the SAD dataset, and does not include the cardiac muscle scene used in the SAD paper. Although this raises some concern regarding the robustness of the method, I do not want to place much weight on this point since I did not raise it earlier during the review. I mention it only so that the authors may consider it in future iterations. Even excluding this point, I still do not think the paper makes a strong enough case.

Given these arguments, I believe I would have to lower my rating.

**Key Questions For Authors:**

1. Can you clarify exactly how the focus measure curves in Figs. 2, 3, and 4 are constructed for each compared method? In particular, for methods that produce spatial response maps, is the plotted value computed per pixel, per block, or by averaging over the full image?

**Limitations:**

yes

**Strengths And Weaknesses:**

Strengths:
1. This paper is a technically credible study on autofocus for spike cameras. The main idea, as I understand it, is to exploit a frequency domain cue during a focus sweep and turn it into a practical autofocus method. Frequency domain analysis itself is not new, but its use here for spike camera autofocus is reasonably motivated, and the paper is generally careful about the setting it tries to solve.
2. The proposed formulation is simple, interpretable, and empirically effective in the target setting.

Main Concern:

While this is not necessarily a weakness of the paper itself, I have some concern about venue fit. As far as I can tell, the paper does not involve a learning component, and the contribution is primarily a handcrafted frequency domain autofocus method for spike cameras. For that reason, the work seems more naturally suited to a computer vision venue than to ICML.

Weaknesses:
1. The empirical validation is relatively narrow and appears to be centered mainly on the SAD benchmark. Because of this, the paper supports the claim that the method works well in this specific setting more strongly than the broader claim that it generalizes across spike camera autofocus scenarios.
2. The comparison could be stronger by including more established and stronger focus measures from the intensity image autofocus literature. Even if such methods are designed for conventional cameras, testing them on spike data would make the comparison more informative. For example, RDF is proposed in [1], and another focus measure, DRDF, is introduced in [2]. Since the code for both methods appears to be available, they could potentially be included as additional baselines.


Question for the authors:
It is not fully clear how the focus measure curves shown in Figs. 2, 3, and 4 are computed. Are these curves obtained per pixel, per block, or by averaging responses over the whole image? For example, for Laplacian-based methods, the operator gives a spatial response map, so how is the final scalar value at each focus step obtained for plotting the curve?


[1] Jeon, Hae-Gon, et al. "Ring difference filter for fast and noise robust depth from focus." IEEE Transactions on Image Processing 29 (2019): 1045-1060.

[2] Ashfaq, Khurram, and Muhammad Tariq Mahmood. "A dual-stage focus measure for vector-valued images in shape from focus." Pattern Recognition 170 (2026): 112112.

---

> ### Author Rebuttal · Authors · 2026-03-27
>
> We thank the reviewer for the careful reading and constructive feedback.
>
> ---
> ## **Main concern: Venue fit**
>
> We thank the reviewer for this important point.
>
> Our work targets the identification of a sensor-observable structure, rather than a task-specific autofocus operator. Specifically, we show that spectral energy migration during a focus sweep provides a stable statistic that encodes focus state in spike measurements, forming a frequency-domain representation.
>
> From this perspective, the contribution is to extracting a meaningful and stable statistic from data. This aligns with a line of work in ICML that focuses on structure extraction and representation, such as spectral approaches that encode intrinsic geometry through frequency-domain statistics (e.g., Proto-value functions in ICML 2005).
>
> Moreover, the formulation is compatible with learning-based approaches: the spectral-migration signal can serve as an inductive bias or supervisory signal for future learning frameworks on neuromorphic data.
>
> We will clarify this positioning in the revision.
>
> ---
>
> ## **Weakness 1: Limited evaluation**
>
> We thank the reviewer for raising the concern about generalization.
>
> Our goal here is to validate the stability of the spectral-migration cue, rather than to exhaustively cover all spike-camera configurations.
>
> While evaluated on the SAD benchmark, this dataset already includes diverse scenes with varying motion and illumination (Tables 3–4). The consistent behavior of the response curves indicates that the proposed cue remains stable across different conditions within the focus sweep.
>
> Importantly, the formulation relies on a sensor-observable frequency-domain property induced by defocus, rather than dataset-specific tuning, which supports its applicability beyond a particular dataset.
>
> ---
>
> ## **Weakness 2: Missing comparisons**
>
> We thank the reviewer for this constructive suggestion. Following this advice, we include additional comparisons with RDF [1] and DRDF [2] and additionally consider two classical focus measures, gradient (GRAD) and Laplacian (LAP).
>
> To enable evaluation of these image-based methods on spike data, we accumulated spike blocks using two representative reconstruction approaches, TFP and TFI [3-4], and compute scalar focus scores accordingly.
>
> **Results (Mean AbsErr (RelErr, %) ↓):**
>
> | Scene  | TFP-GRAD | TFP-LAP | TFP-RDF | TFP-DRDF | TFI-GRAD | TFI-LAP | TFI-RDF | TFI-DRDF | CEN |
> |--------|----------|---------|---------|----------|----------|---------|---------|----------|-----|
> | Bottle | **0.50 (0.3)** | 1.50 (1.0) | 2.00 (1.4) | 2.00 (1.4) | _1.00 (0.7)_ | _1.00 (0.7)_ | 4.50 (3.0) | 3.50 (2.3) | _1.00 (0.7)_ |
> | Lily   | 13.50 (3.8) | 330.50 (92.7) | 14.50 (4.1) | 17.00 (4.7) | 34.50 (9.6) | 340.00 (95.0) | 150.00 (42.8) | _11.50 (3.3)_ | **2.50 (0.7)** |
> | Fan    | _3.50 (3.1)_ | 88.50 (89.5) | 4.50 (4.4) | 52.00 (43.3) | 89.50 (87.0) | 89.50 (87.0) | 91.00 (92.0) | 91.00 (92.0) | **2.00 (1.9)** |
> | USAF   | _43.11 (6.7)_ | 550.11 (91.6) | 308.78 (50.2) | 447.11 (70.2) | 95.22 (14.1) | 138.78 (19.6) | 85.56 (12.7) | 85.56 (12.7) | **3.44 (0.5)** |
> | **Mean** | _28.20 (5.0)_ | 386.13 (79.4) | 188.07 (31.4) | 277.73 (48.7) | 73.80 (21.4) | 140.67 (36.1) | 84.07 (26.0) | 65.47 (20.7) | **2.80 (0.8)** |
>
> RDF and DRDF are well-designed image-based autofocus methods that improve robustness to noise and structural variations, and they provide strong reference baselines in this setting. However, their performance here is influenced by the quality of the intermediate reconstruction, which may introduce smoothing and temporal aggregation artifacts.
>
> In contrast, CEN operates directly in the spike domain and evaluates a frequency-domain cue, thereby avoiding reconstruction-induced information loss and achieving more stable performance.
>
> We will include these comparisons and clarify this discussion in the revision.
>
> **References**
> [3] A retina-inspired sampling method for visual texture reconstruction, ICME, 2019.
>
> [4] Ultra-high temporal resolution visual reconstruction from a fovea-like spike camera via spiking neuron model, TPAMI, 2023.
>
> ---
>
> ## **Key question: How are focus curves constructed?**
>
> We thank the reviewer for the helpful comment.
>
> The focus-response curves in Figs. 2–4 are computed at the **block level**, as indicated in the figure captions. Specifically, spike frames are first accumulated into blocks (Eq. (1)), and each method produces a scalar focus score per block.
>
> For methods that output spatial response maps (e.g., GRAD, LAP), the scalar score is obtained by **averaging over all pixels** within each block. For temporal or frequency-based methods, the score is directly computed from block-level statistics.
>
> To improve clarity, we will:
> (1) explicitly state block-level evaluation;
> (2) enlarge and label axes in Figs. 2–4;
> (3) clarify score computation for each baseline.

---

> > ### Author Rebuttal · Reviewer_KTXp · 2026-04-04
> >
> > I thank the authors for their detailed response and clarifications.  That said, my two main concerns remain unaddressed, so I will keep my current scores and leave the final judgment to the ACs and PCs.
> >
> > 1. **Venue fit**- My main concern regarding venue fit is still unresolved. I did not find the authors' response fully convincing on this point. The rebuttal mainly reframes the contribution as extracting a meaningful and stable statistic from data, but this alone does not clearly establish a connection to machine learning. Many works across different areas extract meaningful statistics or design effective algorithms, but that by itself does not make them a strong fit for an ML venue. The comparison to Proto Value Functions also did not fully address my concern. The relevant mentioned paper's full title is **Proto-Value Functions: Developmental Reinforcement Learning -ICML(2005)**. As I understand it, that work is directly tied to reinforcement learning and representation learning (do correct me if the authors are referring to another ICML paper), whereas the present paper appears primarily as a handcrafted autofocus method for spike cameras. In its current form, I do not yet see a clear machine learning component, nor a strong benefit to the machine learning community specifically. For this reason, I still feel that the work may be better suited to a vision focused venue. I will leave this aspect to the ACs and PCs to judge.
> >
> > 2. **Limited evaluation**- My concern about the evaluation scope also remains. The empirical validation is still limited to a single dataset and a relatively narrow set of scenes. For a venue at the ICML level, I would have liked to see a broader experimental scope.
> >
> > 3. **Additional baselines**-  I appreciate the authors for including additional measures such as RDF and DRDF. This makes the paper appear better than in the original submission. While GRAD and LAP are classical and somewhat old baselines, the inclusion of more recent measures is helpful and makes the evaluation more informative.
> >
> > 4. **Clarification on focus curves**- The authors' clarification regarding the focus response curves is helpful. Computing the curves at the block level seems reasonable. In my experience, I do note that, in practice, averaging within blocks can make gradient/derivative-based methods such as GRAD, LAP, RDF, and DRDF more susceptible to noise. However, I think this issue would likely remain challenging even under pixel-level computation, so I consider the authors' clarification acceptable.
> >
> > Overall, I appreciate the authors' rebuttal and the effort they made to improve the paper.  However, my two main concerns, especially venue fit and also the limited evaluation scope, still remain. Therefore, I will keep my current scores and leave the final decision to the ACs and PCs.
> >
> > I wish the authors the best.

---

> > > ### Author Response · Authors · 2026-04-07
> > >
> > > We thank the reviewer for the careful follow-up and for the thoughtful assessment.
> > >
> > > ## Venue fit
> > > We agree that our method is not a learning-based model in the narrow sense. The main contribution is to identify a structured signal in spikes for focus estimation. As noted by reviewer rwQi, spike-based focus perception is a direction of growing interest in neuromorphic learning, and our work studies a sensing modality that is related to representation learning and application-driven machine learning.
> > >
> > > To address the reviewer’s question on the relevance to learning, we add a supplementary controlled experiment. This experiment is **only provided as additional evidence in response, and does not change the scope of the paper.**
> > >
> > > Specifically, we train the same student model with the same dataset (SAD) and the same training protocol, and only vary the supervision source: (1) a generic sweep-center prior, and (2) the proposed CEN-derived pseudo supervision.
> > >
> > > The student model is a small MLP with two hidden layers (128 units each, dropout 0.1), taking 64-dimensional radial spectral features as input. Training uses a pairwise ranking loss, Adam optimizer (learning rate $10^{−3}$, weight decay $10^{−4}$), batch size 128, and 20 epochs.
> > >
> > > **Table 1. Effect of CEN supervision on learning model.**
> > > | Scene  | Center Err ↓ | CEN Err ↓ | Gain ↑ |
> > > |--------|--------------|-----------|--------|
> > > | Bottle | 35.5         | 14.5      | +21.0  |
> > > | Fan    | 0.5          | 0.5       | 0      |
> > > | Lily   | 44.0         | 30.0      | +14.0  |
> > > | USAF   | 31.9         | 25.7      | +6.2   |
> > > | **MEAN** | **31.9**   | **20.8**  | **+11.1** |
> > >
> > > As shown in Table 1, replacing the center prior with CEN supervision reduces the focus error across scenes (≈35%).  This result suggests that the signal identified by CEN is not only useful for the direct focus estimation task studied in this paper, but can also serve as a more informative supervisory cue in a controlled learning setting.
> > >
> > > In this sense, we believe the work aligns with the [ICML 2026 Call for Papers](https://icml.cc/Conferences/2026/CallForPapers), which explicitly includes “application-driven machine learning” and “representation learning”, as discussed above.
> > >
> > > ---
> > > ## Evaluation scope
> > > We agree that broader evaluation would strengthen the paper. Since SAD is the **only** available dataset for spikes autofocus, we conducted **additional experiments on synthetic focus sweep data in response to the reviewer’s concern.**
> > >
> > > We use synthetic focus sweep sequences from the [CVIA dataset](https://sites.google.com/view/cvia/downloads?authuser=0). Each image is converted into spike sequences following [Noise-Modeled Diffusion Models for Low-Light Spike Image Restoration, ICCV 2025].
> > >
> > > We follow the same evaluation protocol as the main paper, and **keep the CEN parameters identical to the SAD dataset to demonstrate generalization and robustness.**
> > >
> > > **Table 2. Focus estimation error (meters) under moderate-light (m) and low-light (l) conditions**
> > > | Method           | simu01 (m) | simu02 (m) | simu03 (m) | Mean (m) | simu01 (l) | simu02 (l) | simu03 (l) | Mean (l) |
> > > |------------------|------------------|------------------|------------------|------------------|----------------|----------------|----------------|--------------|
> > > | GRAD             | 0.0037           | 0                | 0.0063           | 0.0033           | 0.040          | 0.040          | 0.063          | 0.048        |
> > > | HF_ENERGY        | 0.0037           | 0.0037           | 0                | 0.0024           | 0.0037         | 0.0037         | 0.050          | 0.019        |
> > > | LAPLACIAN_ENERGY | 0.0037           | 0.022            | 0                | 0.0086           | 0.040          | 0.040          | 0.056          | 0.046        |
> > > | MFDCT_FILTER     | 0.0037           | 0.0037           | 0                | 0.0024           | 0.040          | 0.040          | 0.056          | 0.046        |
> > > | SD               | 0.0037           | 0.0037           | 0                | 0.0024           | 0.12           | 0.022          | 0.063          | 0.069        |
> > > | SPIKE_COUNT      | 0.040            | 0.040            | 0.0063           | 0.029            | 0.12           | 0.066          | 0.025          | 0.072        |
> > > | **OURS_CEN**     | **0**            | **0**            | **0**            | **0**            | **0.029**      | **0.0037**     | **0.0063**     | **0.013**    |
> > >
> > > As shown in Table 2, under moderate-light (m), most baselines still produce reasonable responses since structural information is preserved, while CEN remains the most consistent. Under low-light (l), spike signals become sparse and noisy, leading to unstable focus response for local-gradient and high-frequency methods. In contrast, CEN maintains a clear and stable focus-centered cue, resulting in more reliable focus estimation.
> > >
> > > ---
> > > We sincerely thank the reviewer again for the thoughtful and constructive feedback. We also wish the reviewer all the best.

---

### Decision · Program_Chairs · 2026-04-30

**Decision:**

Accept (regular)

**Comment:**

This paper proposed a frequency-domain method based on spectral-centroid migration during a focus sweep. Overall,  the problem is important and the method simple, interpretable, and effective for the target setting.

The main concerns were the limited evaluation scope to a single benchmark (KTXp, vsPL), the heuristic nature of some design choices and the need for stronger theoretical support (rwQi, vsPL, AzEy), and the practical overhead of FFT-based processing for edge deployment (AzEy). However, the rebuttal addressed most of these issues. The authors added stronger baselines, including reconstruction-based pipelines, provided parameter sensitivity analysis and additional theoretical justification, and committed to explicitly discussing boundary-failure sensitivity and FFT-related deployment costs in the final version.

Overall, the balance of reviewer opinion is positive, and the AC finds that the paper makes a technically solid contribution to spike-camera autofocus despite some remaining limitations, and recommends acceptance.